# Mercury Toxicity and Detection Using Chromo-Fluorogenic Chemosensors

**DOI:** 10.3390/ph14020123

**Published:** 2021-02-05

**Authors:** Vinita Bhardwaj, Valeria M. Nurchi, Suban K. Sahoo

**Affiliations:** 1Department of Chemistry, Sardar Vallabhbhai National Institute of Technology (SVNIT), Surat 395007, India; bhardwajvinita1@gmail.com; 2Dipartimento di Scienze della Vita e dell’Ambiente, Università di Cagliari, Cittadella Universitaria, 09042 Monserrato-Cagliari, Italy

**Keywords:** fluorescent sensors, colorimetric sensors, mercuric ions, mercury toxicity

## Abstract

Mercury (Hg), this non-essential heavy metal released from both industrial and natural sources entered into living bodies, and cause grievous detrimental effects to the human health and ecosystem. The monitoring of Hg^2+^ excessive accumulation can be beneficial to fight against the risk associated with mercury toxicity to living systems. Therefore, there is an emergent need of novel and facile analytical approaches for the monitoring of mercury levels in various environmental, industrial, and biological samples. The chromo-fluorogenic chemosensors possess the attractive analytical parameters of low-cost, enhanced detection ability with high sensitivity, simplicity, rapid on-site monitoring ability, etc. This review was narrated to summarize the mercuric ion selective chromo-fluorogenic chemosensors reported in the year 2020. The design of sensors, mechanisms, fluorophores used, analytical performance, etc. are summarized and discussed.

## 1. Introduction

Metals like Na, K, Mg, Ca, V, Cr, Mn, Fe, Co, Ni, Cu, Zn, and Mo are well-known to play important roles in human physiological functions. However, the excessive as well as deficiency of these essential metals in human body can cause serious detrimental effects. Similarly, several non-essential metals entered into the human body from different sources can also cause grievous toxic effects even at trace quantity. Therefore, there is an exponential growth in the development of facile and cost-effective analytical techniques for the on-site and real-time detection of both essential and non-essential metal ions [1,2,3,4,5]. Among the various analytical techniques, the chromogenic and fluorogenic chemosensors are extensively developed for the detection of metal ions because of their high selectivity and sensitivity, easy-to-design, low-cost, simplicity, real-time, and on-site detection ability. The chromo-fluorogenic chemosensors are designed and developed by considering three important things: (i) signaling unit, (ii) recognition unit, and (iii) mechanism (Figure 1). The signaling unit may be an organic fluorophore, chromophore, or optically active nanoparticles. When the recognition unit selectively recognize the target analyte, the mechanism based on electron/charge/energy transfer occurred in the sensor can alter the electronic properties of the signaling unit that gives detectable optical response [6].

Mercury (Hg), one of the non-essential heavy metal can cause serious toxicity to human health and ecosystem. Because of the high affinity to S-containing ligands, the accumulation of mercury in human body can affect the normal functioning of proteins and enzymes leading to the wide variety of diseases related to kidney, brain, reproductive disturbance, central nervous system, etc. [7]. Considering the toxicity, the acceptable limit of inorganic mercury in drinking water was prescribed as 2 μg L^−1^ (10 nM) by the United States Environmental Protection Agency (US EPA). Also, the inorganic mercury can be converted into organic mercury (like methylmercury) that affects the brain and cause other neurotoxic effects, and therefore, the intake of 1.6 µg/kg body weight per week of methylmercury was recommended by Joint FAO/WHO Expert Committee on Food Additives (JECFA) [8]. The associated toxicity even at a trace amount of mercury resulted an expedite growth in the design of novel analytical methods, including optically active chemosensors for the detection of mercuric ions. Literature survey supported the reports of several reviews on mercuric ions sensing and toxicity [9,10,11,12,13,14,15]. In this review, the mercury toxicity and important chelates available for mercury intoxication will be discussed first, and then the chemosensors reported in the year 2020 will be summarized. The fluorophores used for the designing of sensors, the sensing mechanisms and the detection performance will be summarized and discussed.

## 2. Mercury Toxicity and Intoxication

Mercury, a silvery colored metal, liquid at room temperature, is characterized by atomic number 80 in group 12 of the periodic table of elements, standard atomic weight 200.59 g/mol. Mercury can assume the three oxidation states 0, +1, +2. It presents a high density 13.53 g/mL, and a relatively high vapor pressure (0.0017 torr at 25 °C, corresponding to a concentration of 20 mg/m^3^). It is monoatomic in vapor phase, and is highly soluble in polar and non-polar solvents (a mercury water solution can reach the concentration of 0.6 μg/L at 25 °C).

The use of mercury is reported since the ancient times, mainly as the pigment cinnabar. The mined amount of mercury has been almost constant over the centuries until 1500, when huge amounts were produced in Spain (Almaden) to be shipped to Spanish South America for silver extraction. A paper by Hylander and Meili [16], takes into account the trend in mercury production from this period to 2000. The discovery of gold in California in 1850 produced a jump in mercury production, as well as its use in chlor-alkali plants in the 20th century. The consumption of explosives in the war industry contributed to the large production of mercury during World Wars I and II. The increasing awareness of mercury toxicity has led in the years to its banning from different applications (amalgamation in China in 1985 and in Russia in 1990, pesticides in USA in 1993, batteries in USA in 1996), until the *Protocol on Heavy Metals* (cadmium, lead and mercury), signed in 1998 by different countries, the 2005 *EU Mercury Strategy*, and finally the *Minamata Convention on Mercury* in 2013. At the fifth session of the Intergovernmental Negotiating Committee in Geneva, Switzerland, on 19 January 2013, it was agreed the *Minamata Convention on Mercury*, a global treaty to protect human health and the environment from the adverse effects of mercury. The major highlights of the Minamata Convention on Mercury include a ban on new mercury mines, the phase-out of existing ones, control measures on air emissions, and the international regulation of the informal sector for artisanal and small-scale gold mining [17]. Despite the above legislative controls, mercury remains one of the major toxicants in the world [18] and deserves a careful consideration about its environmental quantification, its toxic action, and the strategy for the clinical treatment of intoxication.

Mercury presents in the environment mainly in three chemical forms, i.e., elemental mercury (liquid or vapor Hg^0^), inorganic mercuric compounds (Hg^2+^), and organic mercury compounds (methylmercury, MeHg, CH_3_Hg and ethyl mercury EtHg, C_2_H_5_Hg) [19]. Toxicity of mercury in humans can be related to any of these three forms, absorbed in different ways: inhalation, oral, and dermal. The kind and the degree of intoxication is highly specific for any of these three chemical species, as well as the symptoms and the consequences [20]. Table 1 presents the main sources of exposure of the different forms of mercury and the affected organs [21].

The exposure of mercury by human body can be occurred via ingestion or inhalation [7]. The extensive utilization of elemental mercury in a number of industrial processes has led the involved personnel exposed to gaseous mercury. To have a quantitative evaluation of this exposure, we remember that in presence of liquid Hg^0^ the surrounding non-ventilated air can reach a concentration of 20 mg/m^3^ of monoatomic mercury vapors. Since a person inhales 15–20 m^3^ of air daily, a worker who stays about 8 h in a mercury saturated place inhales 5–6.6 m^3^ of air, i.e., 100–135 mg of Hg^0^. Mercury vapors are efficiently absorbed by lungs due to their high liposolubility producing severe pulmonary injuries. Elemental mercury in the lungs enters the blood flow, where a certain amount is oxidized to Hg^2+^ and excreted in urine, and some, due its high liposolubility, passes through the blood–brain barrier (BBB) and enters in the central nervous system.

Various forms of inorganic mercury in water are converted by microorganisms to methyl mercury that accumulates in fish and pass to humans through the food chain. In humans, inhaled or ingested methyl mercury is well absorbed and is found in red blood cells, liver, kidneys, and above all in the brain (including the fetal brain, since methyl mercury can cross the placental barrier), where it causes severe, cumulative, and irreversible injuries to the central nervous system. Its retention time in the human body varies from months to years, and the appearance of symptoms can be delayed for many years. Symptoms of methylmercury intoxication include mental retardation, cerebral blindness, deafness, palsy, and dysarthria, particularly in children exposed in utero. It is important to emphasize that methylmercury exposure mainly affects people whose diet includes the consumption of high amounts of fish.

Inorganic mercury compounds were largely used in the chemical industry, and were the cause of heavy occupational exposure. Inorganic salts are poorly absorbed, and kidneys represent their main target. From a chemical point of view, mercury toxicity depends primarily from the mercuric ion ability to form covalent bonds with sulfur atoms, substituting hydrogen atoms in sulfhydryl groups of proteins to form mercaptides. This can deactivate a number of essential enzymes, completely altering their regular biological functions [21,22,23].

Chelation therapy is used for the treatment of all forms of mercury intoxication. In clinical use, chelating agents remove metal ions from the biological ligands in the organism, where they exert their toxic action, through the formation of metal complexes that are successively excreted. Characteristics of a good chelator should be great solubility in both water and lipids, resistance to biotransformation, capacity to reach the sites of metal accumulation, high stability of the complexes at the pH of body fluids, and toxicity of the formed complexes lower than that of the free metal ions [21]. Most of the chelating agents in use today are not able to cross the BBB and this limits their ability to remove the toxic metal ions from the brain. The main chelating agents used in the treatment of mercury intoxication are summarized in Figure 2.

The dithiol chelating agent 2,3-dimercaptopropan-1-ol (BAL) was originally synthesized for the treatment of the effects of the war gas Lewisite. It competes successfully with protein SH groups forming stable chelates with Hg^2+^ metal ions. For several decades after its synthesis, it was recommended for the treatment of inorganic mercury poisoning, but it presents severe adverse effects, including painful intramuscular injections, high blood pressure and tachycardia, and predisposition to redistribute the complexed toxic elements to the brain. At any rate, it is contraindicated in the treatment of alkyl-Hg intoxication. In most cases, it has been replaced by DMSA and DMPS in the treatment of metal poisoning [24].

The chelating agent *meso*-dimercaptosuccinic acid (DMSA) or simply called succimer is the water-soluble dithiol. DMSA can be administered as intravenous and oral preparations, being DMSA a hydrophilic chelator. When administered orally, about 20% is absorbed in the gut, and about 95% of the absorbed amount is bound to plasma albumin, presumably by one SH group to a cysteine residue, being the second SH group free for binding metal ions. The action of DMSA is limited to the extracellular space. It increases Hg excretion in the urine. DMSA is considered the drug of choice for the treatment of organic-Hg intoxication. Even if DMSA does not pass the BBB, it indirectly reduces the brain burden of methyl mercury presumably by changing the brain/blood equilibrium. The derivative monoisoamyl 2,3-dimercaptosuccinic acid (MiADMSA) is at the moment under evaluation. Differently from DMSA, which removes extracellularly distributed metal ions [25], MiADMSA is also able to chelate intracellular distributed metal ions [26].

The structure of 2,3-dimercaptopropane-1-sulfonic acid (DMPS), also known as unitiol is shown in Figure 2. DMPS is a drug produced in Germany and registered for the treatment of mercury intoxication. It is not an approved drug in the USA, unless the FDA gives a special permission. The daily dose is usually 3–10 mg DMPS/kg body weight. DMPS is believed the optimal remedy in poisoning by inorganic mercury [27], while it is less effective than DMSA for organic mercury [28]. DMPS can be administered both intravenously and orally; less than 40% of orally administered DMPS is effectively adsorbed [29]. DMPS, which is a hydrophilic chelating agent, is mainly distributed in the extracellular space, but a small fraction is found in the intracellular compartments [30]. DMPS scavenges mercury from kidneys more effectively than DMSA, and is considered the drug of choice for the treatment of acute intoxication by inorganic mercury [30,31].

The α-lipoic acid ((*R*)-5-(1,2-dithiolan-3-yl)pentanoic acid, LA) is the sulfur containing organic compound known as thioctic acid, presented in Figure 2. LA, essential for aerobic metabolism, is ordinarily produced in the body. Its reduced form, called dihydrolipoic acid (DHLA), contains a couple of -SH groups; it is characterized by high affinity for mercuric ion and has been recently proposed as an effective mercury chelator [31].

## 3. Chromo-Fluorogenic Chemosensors

Because of the potential toxicity of mercuric ions to living systems, there is an expedite growth in the design of optically activity chemosensors. In the year 2020, more than 100 Hg^2+^ selective chromo-fluorogenic chemosensors were reported, which can be classified in three different categories based on the optical responses, i.e., (i) fluorescence, (ii) colorimetric, and (iii) dual-mode chemosensors. The fluorescence chemosensors are discussed based on the fluorescence changes, i.e., turn-off, turn-on and ratiometric (Table 2). The fluorescent chemosensors are more sensitive than the colorimetric sensors with high visual effects that allowed for different bioimaging and diagnosis applications. The most dominating mechanisms for Hg^2+^ sensing are complexation-induced change in the optical properties due to electron/energy/charge transfer or the sensor possess a reactive group that undergoes Hg^2+^-catalyzed chemical transformation. The sensing mechanisms and other experimental parameters (such as solvent medium, pH and concentration of sensors etc.) important in fabricating a suitable chemosensors are discussed.

### 3.1. Fluorescent Chemosensors

#### 3.1.1. Fluorescent Turn-Off Chemosensors

The heavy metal ions like Hg^2+^ greatly influence the fluorescence of a sensor after complexation leading to the fluorescence quenching via energy or electron transfer mechanism. Ebru et al. [32] have reported the pyrazoline based fluorogenic sensor **1** for the detection of Hg^2+^ in aqueous medium. Sensor **1** (Figure 3) showed a fluorescence maxima at 464 nm (λ_exc_ = 350 nm), but the fluorescence intensity was decreased upon addition of Hg^2+^ with the sensitivity limit of 0.16 μM (Figure 4). The electrostatic interaction between **1** and the heavy metal ion Hg^2+^ caused the fluorescence quenching. This reversibility sensor **1** formed a complex with Hg^2+^ in 2:1 stoichiometry. The fluorescent turn-off sensor **2** was introduced for the detection of Hg^2+^ in water medium [33]. Sensor **2** (Figure 3) showed an absorption band at 525 nm while fluorescence maxima at 632 nm. Prominent fluorescence quenching accompanied by 25 nm red-shift was observed upon binding with Hg^2+^ leading to the solution color turned from pink to pale violet due to the intramolecular charge transfer occurred between Hg^2+^ and the N-atoms of **2**. Sensor **2** formed complex with Hg^2+^ in 1:2 stoichiometry, and the estimated LOD was reported to be 39.2 nM.

The Hg^2+^ selective fluorescent turn-off sensor **3** (Figure 3) using the Zn-based metal organic framework (Zn-MOF) was prepared by reacting the ligand 5-aminoisophthalic acid with Zn^2+^ [34]. The Zn-MOF formed a 3D supramolecular network having uncoordinated carboxylic atoms and pores size of 8.2 Å. Addition of Hg^2+^, the fluorescence of Zn-MOF at 416 nm (λ_exc_ = 316 nm) was quenched with a sensitivity limit of 0.1243 μM due to the complexation-induced inhibition of intermolecular energy transfer. In another work, the triarylamine-based covalent organic framework (COF) polymer **4** (Figure 3) was converted into nanosphere via Suzuki polymerization under mini-emulsion condition, which showed selective fluorescence turn-off response for Hg^2+^ in mixed aqueous medium. The blue-green fluorescence of **4** is quenched upon complexation of Hg^2+^ with the sulfur atom. The sensor was immobilized successfully over macroporous sponge for facile detection and removal of Hg^2+^ [35].

The benzimidazole derived fluorescent sensor **5** (Figure 5) showed an excellent selectivity towards Hg^2+^ in CH_3_CN/H_2_O (1:1, *v*/*v*). Sensor **5** exhibited a fluorescence emission at 380 nm when excited at 270 nm. In the fluorescence experiments, only Hg^2+^ caused significant fluorescence quenching (85%) of **5** by forming a complex in 1:1 stoichiometry (Figure 6). Sensor **5** showed a LOD of 0.68 μM, and no interference with other tested metal ions. The complex [**5**-Hg^2+^] emits in an acidic environment whereas quenched in an alkaline environment, which can also be used for pH sensing [36]. The triazole-bridged coumarin conjugated quinoline sensor **6** (Figure 5) was developed for the fluorescent turn-off detection of Hg^2+^. The complexation of **6** with Hg^2+^ at the tridentate coordination site created by the quinolone and triazole stimulates the unusual PET process, and caused the fluorescence quenching at 485 nm with the sensitivity limit of 172 nM. The Hg^2+^ sensing ability of **6** was further studied in live U-2-OS cells [37]. Murugan et al. [38] have reported the tetraazamacrocyclic derivative appended with the salicylaldehyde **7** (Figure 5) for the selective fluorescent turn-off sensing of Hg^2+^ in CH_3_CN/HEPES buffer (2:8, *v*/*v*). The fluorescence emission of **7** at 490 nm was quenched upon complexation with Hg^2+^ in 1:1 ratio due to chelation enhancement quenching (CHEQ) effect. The quenched fluorescence is recovered with the addition of KI. The lowest limit of detection for Hg^2+^ is 1 nM. In addition, the sensor **7** showed fluorescence turn-on response for the detection of HSO_4_^−^.

The marine cyanobacterium based natural protein C-phycoerythrin **8** (Figure 5) was applied for the fluorescent turn-off detection of Hg^2+^. Sensor **8** showed an intense yellow orange fluorescence at 574 nm due to the phycoerythrobilin (PEB), a linear tetrapyrrole. Upon interaction of Hg^2+^ with the amino acid side chain and thioether bridges in the protein **8**, the fluorescence is quenched. The complexation of Hg^2+^ caused indirect charge transfer that quenched the fluorescence of **8**. The LOD of **8** for Hg^2+^ was estimated as 312 nM [39].

The acridine-based chemosensor **9** (Figure 7) possessing two S-donor atoms was developed for the fluorescent turn-off detection of Hg^2+^ in Tris-HCl buffer. The fluorescence of **9** at 445 nm was quenched upon complexation with Hg^2+^ in 1:1 binding stoichiometry. Sensor **9** showed a LOD of 4.40 μM, and applied for the monitoring of Hg^2+^ in real water samples and bioimaging ability in living cells [40]. Adopting the complexation-induced fluorescence quenching approach, the dansyl-peptide based sensor **10** was developed by the conjugating two serines and dansyl groups. Sensor **10** (Figure 7) exhibited sensitivity towards Hg^2+^ through fluorescence quenching at 550 nm in HEPES buffer solutions. Upon complexation of sensor **10** with Hg^2+^ in 2:1 stoichiometry, the heavy atom effect and the electron transfer caused the fluorescence quenching. With nanomolar detection limit (7.59 nM), sensor **10** was successfully applied for monitoring Hg^2+^ ions in real water samples (lake and tap water) and living LNCaP cells [41].

Thiocarbohydrazide based Schiff base **11** (Figure 7) was introduced for the colorimetric and fluorescent sensing of Hg^2+^. Sensor **11** showed AIE behaviour in a mixture of acetonitrile and water. The emission intensity was found to increase gradually with the addition of water up to 40%, and the cyan fluorescence was clearly developed from aggregates. Upon interaction of Hg^2+^ with the AIE active **11** led to the color change from colorless to yellow. The quenching in fluorescence intensity was attributed to combine effect of chelation enhanced fluorescence quenching (CHEQ) and photo-induced electron transfer (PET). The limit of detection for Hg^2+^ is 1.26 nM. Sensor **11** was applied for the detection of Hg^2+^ by using test paper strip and in various real water samples [42].

Yanxin et al. [43] reported a covalent organic frameworks (COFs) **12** (Figure 7) with extended hydrazone-linked π-conjugation by condensing two different monomers for the detection and removal of Hg^2+^ in acetonitrile. Sensor **12** showed an absorption peak at 350 nm while emission band at 603 nm due to the ESIPT. Addition of Hg^2+^ ion, the color change from orange to light blue with the significant fluorescence quenching due to the inhibition of the ESIPT process. Limit of detection is calculated as 20 ppb without any significant interference with other ions. Moreover, sensor **12** was applied for the effective Hg^2+^ removal from water. The pyrene-based COFs **13** (Figure 7) was introduced for the simultaneous detection and removal of Hg^2+^ in DMF. After interaction with Hg^2+^, the fluorescence of COFs **13** at 463 nm was quenched with the sensitivity limit of 17 nM, and the blue-emitting **13** turned to colorless. Fluorescence quenching of **13** is attributed to a PET process from sensor to the Hg^2+^. Sensor **50** was applied for removing Hg^2+^ from both air and water [44]. The microporous porphyrinic zirconium-based MOF **14** was developed by using meso-tetra(4-carboxyphenyl)porphyrin as a ligand for the detection of Hg^2+^ in methanol medium [45]. Sensor **14** fluorescence at 436 nm was quenched upon addition of Hg^2+^ ions with a fast response rate under <1 min and sensitivity limit of 0.01 μM. The quenching efficiency was explain by donor-acceptor (D-A) electron transfer mechanism. Also, sensor was applied for the detection of DMF.

Reena et al. [46] have reported a phenylalaninol-fluorescein conjugated Schiff base receptor **15** (Figure 8) for the colorimetric and fluorescence detection of Hg^2+^ in pure aqueous medium. Upon gradual addition of Hg^2+^, the emission at 521 nm is quenched and slightly red-shift, while the absorption showed a hypsochromic shift of 30 nm at 430 nm causing color change from green to light pink. The lowest limit of detection for Hg^2+^ is 0.34 μM. The job’s plot supported the 1:1 binding stoichiometry between **15** and Hg^2+^. Sensor **15** is applicable for Hg^2+^ detection in industrial effluents and paper strip visualization with the irreversible mode. The curcumin and *β*-cyclodextrin inclusion complex **16** (Figure 8) was applied for the chromo-fluorogenic sensing of Hg^2+^ in aqueous medium. The supramolecular system **16** complexed with Hg^2+^ after deprotonation of aliphatic hydroxy group caused apparent color change from yellow to colorless [47]. The absorbance of **16** at 482 nm was quenched and blue-shifted to 379 nm. Also, the fluorescence emission at 512 was significantly diminished and blue-shifted to 502 nm. With the fluorescence change, the concentration of Hg^2+^ can be detected down to 5.02 µM, and applied to quantify Hg^2+^ concentration in real water samples. In another work, the fluorescent turn-off sensor **17** was developed for the detection of Hg^2+^ in MeOH/H_2_O (1/4, *v*/*v*) solvent [48]. The emission at 455 nm is red-shifted to 485 nm and quenched with the addition of Hg^2+^ due to the chelation enhancement quenching effect (CHEQ). Without responding with other ions, sensor **17** can be applied to detect Hg^2+^ down to 3.12 nM. In addition, the absorption band of **17** at 292 nm was quenched whereas the band at 337 nm red-shifted to 355 nm. The absorbance and fluorescence changes occurred in **17** after the addition of Hg^2+^ were recovered with the addition of iodide ions.

Ashwani et al. [49] reported an anthrapyridone-based receptor **18** (Figure 8) for the sensing of Hg^2+^ and Cu^2+^ in CH_3_CN. The fluorescence emission at 492 nm of sensor **18** was quenched in presence of Hg^2+^ with the LOD of 200 nM. In UV–vis absorption study, the absorption peak of **18** at 445 nm was quenched and a red-shifted absorption band appeared at 630 nm leading to the color change from yellow-green to green. Similar spectral changes of **18** were also observed with Cu^2+^. The quinoline-based benzimidazole derivative **19** (Figure 8) was developed for the dual-mode chemosensing of Hg^2+^ in DMF/H_2_O [50]. Sensor **19** formed gel in DMF and converted to sol with the addition of Hg^2+^, and also the fluorescence at 378 nm was quenched. This gel-sol transition based sensor showed the 16 nM LOD for Hg^2+^. Sensor **19** also showed naked-eye detectable color change for Cu^2+^ from white to dark pink in the gel state.

The multi-analytes selective bis-thiosemicarbazone based receptor **20** (Figure 8) was developed for the detection of Hg^2+^, Zn^2+^ and Cd^2+^ in H_2_O:DMSO (95:5 *v*/*v*) [51]. Sensor **20** showed an intense emission at 540 nm, while addition of Hg^2+^, the fluorescence was quenched and red-shifted to 578 nm with the color changed from yellow to reddish-brown (λ_ex_ = 360 nm). Color and fluorescence changes were attributed by intra-ligand fluorescence and influence of coordination of Hg^2+^ to the receptor. The LOD for Hg^2+^ was estimated as 0.51 μM. With Zn^2+^ and Cd^2+^, the fluorescence of **20** was blue-shifted and enhanced respectively at 488 and 470 nm. The tryptophan-based polymer **21** (Figure 8) was reported for the detection of Hg^2+^ and Cu^2+^ [52]. Sensor **21** showed dual emissions at 364 nm and 464 nm with the yellow colored fluorescence in aqueous medium at physiological pH. Upon complexation with Hg^2+^/Cu^2+^, sensor **21** showed significant quenching at 464 nm without any change at 364 nm. The quenching was attributed by PET process from the tryptophan donor to the pyridine acceptor unit. With **21**, the concentration of Hg^2+^ and Cu^2+^ can be detected down to 7.41 nM and 4.94 nM, respective, and applied for the bioimaging of intracellular Hg^2+^/Cu^2+^ in live CP3 cells. The isocoumarin based sensor **22** (Figure 8) was developed for the fluorescent turn-off sensing of Hg^2+^ and Fe^3+^ DMSO/HEPES buffer solution (9/1, *v*/*v*, pH 7.0) [53]. After complexation of Hg^2+^/Fe^3+^ with **22** in 1:2 stoichiometry, significant fluorescence quenching was observed at 455 nm. Sensor **22** showed the sensitivity limit of 8.12 nM and 5.51 nm for Hg^2+^ and Fe^3+^, respectively. Sensor **22** was applied to imaging intracellular Hg^2+^/Fe^3+^ in live HepG2 cells.

#### 3.1.2. Fluorescent Turn-On and Ratiometric Chemosensors

The fluorescent turn-on and ratiometric sensors are more advantageous than fluorescent turn-off sensors for biological applications because of the facile measuring of low-concentration contrast in compared to a ‘dark’ background, reduction in the false positive signals and enhancement in the sensitivity. Therefore, more numbers of Hg^2+^ selective fluorescent turn-on and ratiometric sensors were reported and applied for the monitoring of Hg^2+^ in real samples and bioimaging intracellular Hg^2+^ in live cells. The micellar based Hg^2+^ selective fluorescent turn-on sensor **23** (Figure 9) was developed by organizing the fluorophore 10-methylacridinium perchlorate, sulfur-containing ligand *N*,*N*-bis(2-hydroxyethylthio-1-ethyl)dodecylamine and the surfactant sodium dodecyl sulfate (SDS) [54]. Under micellar condition, the ligand decorated with the SDS formed complex with Hg^2+^ that enhanced the fluorescence of 10-methylacridinium due to the inhibition of PET from the ligand to the excited fluorophore. The turn-on fluorescence is observed at λ_em_ = 495 nm (λ_exc_ = 359 nm) with the limit of detection of 22 nM Hg^2+^. The pyridyl-based sensor **24** (Figure 9) containing multiple binding sites was developed for fluorescent turn-on sensing of Hg^2+^ in aqueous solution with a limit of detection of 0.28 ppb [55]. The broad fluorescence emission spectrum of sensor **24** with maxima at 387 nm showed about 5-fold emission enhancement upon addition of Hg^2+^ due to the complexation-induced inhibition of PET and C=N isomerization. The applicability of sensor **24** was assessed in real water samples. The ESIPT and PET based sensor **25** (Figure 9) was developed for the detection of Hg^2+^ in DMF/HEPES solution (1:1, *v*/*v*) medium [56]. After interaction with Hg^2+^, sensor **25** showed fluorescence enhancement at 495 nm with 180 nm Stokes shift. The complex formation between **25** and Hg^2+^ in 1:1 stoichiometry inhibited both the ESIPT and PET processes (Figure 10), which caused significant fluorescence enhancement. The limit of detection of **25** was 6.45 × 10^–7^ M Hg^2+^ and applied in environmental and biological samples for Hg^2+^ quantification. In addition, the in situ generated complex of **25** with Hg^2+^ was applied as a secondary sensor for the detection of S^2–^.

Two polystyrene solid-phase sensors **26** and **27** (Figure 9) were synthesized with different lengths of the linker [57]. The fluorescence intensity was determined with an excitation wavelength of 401 nm for **26** and 405 nm for **27**. These naphthalimide-piperazine-pyridine based sensors **26** and **27** showed fluorescence enhancement at 520 nm and 525 nm upon the incremental addition of Hg^2+^ in HEPES buffer (pH 7.2), respectively. The detection mechanism involving the Hg^2+^ sensing is chelation-induced inhibition of PET. Sensor **26** showed a higher fluorescence response than **27** with the LOD of 1.01 µM Hg^2+^. Also, sensor **26** was successfully applied to monitor Hg^2+^ in tap water and lake water.

The near-infrared (NIR) fluorescent receptor **28** (Figure 11) containing a donor-acceptor structure was reported for the detection of Hg^2+^ in THF-H_2_O (9:1, *v*/*v*), where the triphenylamine-benzothiadiazole acts as a fluorophore and the rhodanine-3-acetic acid as metal ion recognition unit [58]. The complexation of Hg^2+^ with **28** in 1:1 stoichiometry at the S and O donor atoms of rhodanine-3-acetic acid enhanced the electron-donating ability and blocked the intermolecular charge transfer process, which caused significant fluorescence enhancement at 675 nm with the sensitivity limit of 13.1 nM. This low cytotoxic sensor **28** was applied for the imaging of intracellular Hg^2+^ in live A549 cells and zebrafish larvae. The chemosensor **29** (Figure 11) based on the piperazine derivative was developed for the detection of Hg^2+^ ion in mixed aqueous DMSO. In UV–vis spectral analysis, sensor **29** showed a strong ICT band at 495 nm that significantly blue-shifted (13 nm), while the emission band at 543 nm was greatly enhanced in the presence of Hg^2+^ with LOD of 19.2 nM. The sensing mechanism was explained by the aspect that after addition of Hg^2+^, the electron donating ability of aniline group was reduced that suppressed the PET process in sensor resulting fluorescent turn-on detection of Hg^2+^ (Figure 12A) along with the naked-eyes colorimetric change from orange to yellow. Sensor **29** was also applied for the potential application in paper strip visualization and bioimaging (Figure 12B) [59]. Another PET based sensor **30** (Figure 11) containing NBD fluorophore and thiophene ionophore was applied for the fluorescent turn-on sensing of Hg^2+^. Sensor **30** showed weak emission at 587 nm in CH_3_CN:H_2_O (4:6 *v*/*v*), while the fluorescence enhanced by 50 folds upon complexation with Hg^2+^ in 1:1 binding stoichiometry. The fluorescence enhancement was observed due to complexation-induced inhibition of PET. The limit of detection of 3.9 ppb Hg^2+^ was estimated for sensor **30**, and was applied to detect Hg^2+^ in drinking water, live cells and plant tissues [60].

Xiaobo et al. [61] recently introduced a bismacrocyclic polyamine-based chemosensor **31** (Figure 11) containing two 4-nitro-1,2,3-benzoxa-diazole molecules for the selective detection of Hg^2+^ in CH_3_CN/HEPES (1:9, *v*/*v*). Sensor **31** showed a fluorescence enhancement at 530 nm upon addition of Hg^2+^ with the sensitivity limit of 27 nM. The binding stoichiometry of **31**-Hg^2+^ complex was 1:1 determined by Job’s plot and ES-MS. Sensor **31** was applied to monitor exogenous Hg^2+^ in living HeLa cells. Furthermore, the complex **31**-Hg^2+^ was applied for the detection of glutathione (GSH) in FBS and human serum.

Madhusmita et al. [62] reported a dual-mode sensor **32** (Figure 11) containing a styrylpyridinium dye for the detection of Hg^2+^ in mixed methanol-H_2_O (4:1, *v*/*v*) medium. Orange color solution of **32** turned to colorless under daylight, whereas started emitting yellow-color under UV light after adding of Hg^2+^. The weak fluorescence band of **32** at 590 nm showed a remarkable blue-shift and enhancement at 566 nm with the LOD of 4.8 μM. Complexation of **32** with Hg^2+^ in 1:1 binding stoichiometry inhibited the PET and increase the conformational rigidity that caused the chelation enhanced fluorescence (CHEF) at 590 nm. Sensor **32** was applied for Hg^2+^ detection in test paper strips, bioimaging in *E. coli* DH5-α cells and mimicking INHIBIT molecular logic gate.

Guilin et al. [63] reported a terpyridine-based probe **33** (Figure 11) for detection of Hg^2+^ in aqueous solution. Probe showed the aggregation-induced emission (AIE) property in mixed DMSO/H_2_O mixture. In acidic medium (pH = 2), significant fluorescent enhancement was noticed in presence of Hg^2+^ with the red-shifting from 453 nm to 521 nm. The fluorescence enhancement is due to the complexation of **33** with Hg^2+^ in 2:1 ratio followed by coordination-triggered self-assembly of **33**. Additionally, probe displayed highly efficient removal of Hg^2+^ ions from solution by rapid precipitation. In another approach, the triphenylamine (TPA) based NIR fluorescent sensor **34** was developed for the sensing of Hg^2+^. Sensor **34** (Figure 11) showed AIE properties with the red-emitting fluorescence at 639 nm in DMSO/H_2_O (1:99 *v*/*v*) mixed media. Sensor **34** is weakly emissive in 80% H_2_O-DMSO mixed solvent, but with the addition of Hg^2+^ caused significant fluorescence enhancement with the spectral shift from 600 nm to 639 nm. The strong fluorescence appeared due to the Hg^2+^-directed aggregation of **34** with the sensitivity limit of 30 nM. Sensor **34** was applied for the bio-imaging in HepG-2 cells [64].

Hai-Ling et al. reported a Cu(II)-based three-dimensional zwitterionic MOF and then functionalized with carboxyfluorescein labeled thymine-rich (T-rich) DNA **35** for the sequential detection of Hg^2+^ and biothiols. The non-fluorescent hybrid MOF **35** showed fluorescence enhancement at 518 nm due to the formation of hairpin-like T-Hg^2+^-T structure with the sensitivity limit of 3 nM. The formation of rigid complex, the MOF is separated that recovered the fluorescence of dye. This MOF based sensing approach was applied on the environmental water and serum samples for Hg^2+^ and homocysteine recovery [65].

İnal et al. [66] reported a salicylaldehyde derived sensor **36** (Figure 13) for the determination of Hg^2+^, Zn^2+^, and Cd^2+^ in ethanol-aqueous medium. The formation of **36**-Hg^2+^, **36**-Zn^2+^, and **36**-Cd^2+^ complexes resulted significant fluorescence enhancement at 491, 452, and 474 nm, respectively. Sensor **36** formed complexes with Hg^2+^ and Zn^2+^ in 2:1 ratio whereas with Cd^2+^ in 1:1 ratio. Sensor **36** showed LOD of 270, 750, and 600 nM towards Zn^2+^, Hg^2+^, and Cd^2+^, respectively. In another work, the quinolone-based sensor **37** (Figure 13) was introduced for the detection of Hg^2+^ with the fluorescence method. In presence of Hg^2+^_,_ the weakly emissive **37** at 463 nm undergoes large fluorescence enhancement at 490 nm, and the fluorescent color changed from faint blue to green [67]. The sensing mechanism was attributed to complex formation between **37** and Hg^2+^ which inhibited the PET and the excited-state intramolecular proton-transfer (ESIPT). The detection limit of Hg^2+^ is 2.1 nM. The binding stoichiometry between **37** and Hg^2+^ is 1:1 performed by job’s plot. Sensor **37** gives the reversible response with the addition of NaBH_4_. In addition, sensor **37** showed selective changes in the presence of Cu^2+^.

Xiao et al. [68] reported the boron dipyrromethene (BODIPY) based monomeric **38** and polymeric sensors (**39** and **40**) for the detection of Hg^2+^ in DMF/buffer (8:2, pH = 7.0) (Figure 13). Emission band of **38** at 529 nm was enhanced after addition of Hg^2+^ due to the complexation-induced inhibition of PET with a sensitivity limit of 2.40 μM, and also the color of sensor solution turned from orange to orange-green. Similar to **38**, the polymeric sensors **39** and **40** also showed high selectivity towards Hg^2+^, and their fluorescence enhanced respectively at 621 and 614 nm. The LOD of sensors **39** and **40** to Hg^2+^ was estimated as 2.86 μM and 0.22 μM, respectively. The polymeric sensors **39** and **40** also showed the colorimetric response from colorless to pink after Hg^2+^ addition. These low cytotoxicity sensors **38**–**40** showed good cell permeability and applied successfully to monitor intracellular Hg^2+^ in live A549 cells and zebrafish.

Rhodamines are extensively applied for the designing of fluorescent turn-on sensors, where the colorless and non-fluorescent ring-closed spirolactam form turned to ring-opened form upon interaction with target analyte that caused significant fluorescent enhancement and color change from colorless to an intense color. The complexation-promoted ring-opening of rhodamines is widely used for the designing of many Hg^2+^ selective fluorescent turn-on sensors. With some exceptions, majority of the sensors discussed here showed dual-mode chromo-fluorogenic response but the fluorescence changes of the sensors are discussed because of their high sensitivity than the UV–vis method with possible application in bioimaging. The rhodamine B based fluorescent organic nanoparticles (FONs) **41** (Figure 14) was prepared via the reprecipitation technique. Upon addition of Hg^2+^, the fluorescence of **41** was enhanced at 532 nm (λ_exc_ = 480 nm) due to the chelation-enhanced fluorescence (CHEF) phenomenon that open the spirolactum ring. Sensor **41** with the LOD of 8.619 nM was successfully applied to quantify Hg^2+^ in environmental samples (tap and river water) and for intracellular Hg^2+^ imaging [69]. With the spirolactum ring opening mechanism, several Hg^2+^ selective fluorescent turn-on sensors were reported. The polyacylamide-fluorescein based sensor **42** was reported for the fluorescent turn-on sensing of Hg^2+^ (λ_em_ = 515 nm, λ_ex_ = 460 nm) in PBS buffer (pH = 7.0) [70]. Sensor **42** (Figure 14) showed turn-on fluorescent response to Hg^2+^ due to the complexation-induced opening of spirolactum ring with the detection limit of 0.4 nM. Sensor **42** showed biological compatibility and cell permeability and successfully applied for turn-on fluorescent determination of Hg^2+^ both in aqueous samples (lake and tap water) and living cells.

Kaijie et al. [71] reported the rhodamine-based sensor **43** (Figure 14) for the selective detection of Hg^2+^ in CH_3_CN-HEPES buffer (1:9, *v*/*v*). After addition of Hg^2+^, sensor **43** showed fluorescence enhancement at 580 nm due to the coordination between **43** and Hg^2+^ in 2:1 binding stoichiometry followed by opening of rhodamine spirolactam ring. In addition, the sensor **43** showed colorimetric response for the detection of Cu^2+^. Heng et al. [72] reported a diarylethene and triazole-linked rhodamine B based sensor **44** (Figure 14) for the recognition of Hg^2+^ in DMSO (λ_ex_ = 520 nm, λ_em_ = 606 nm). Sensor **44** showed a 88-fold fluorescence enhancement at 606 nm with the addition of Hg^2+^ due to the complexation-induced opening of the rhodamine-spirolactam ring. This sensor showed a detection limit of 0.13 µM and applied to mimic the INHIBIT logic gate by taking Hg^2+^ and TFA as two molecular inputs.

Jin et al. reported a near-infrared fluorescent sensor **45** (Figure 15) for the selective detection of Hg^2+^ in HEPES buffer (10 mM, pH = 7.4, containing 20% CH_3_CN), where the thiosemicarbazide moiety served as a recognition site [73]. Sensor **45** showed the potential response towards Hg^2+^ by absorption and fluorescence spectra with the detection limit as low as 1.5 nM with fast response times (3 min). Sensor **45** showed fluorescence enhancement at 691 nm with the large Stokes shift (78 nm), while in the absorption spectra, sensor gives the intense absorption at 613 nm after binding with Hg^2+^ in 1:1 stoichiometry leading to the color change from colorless to dark blue was observed. Sensor **45** also applied as an efficient organelle-targeting sensor for Hg^2+^ in mitochondria of living cells imaging. In another work, the same sensor **45** was applied for the sensing of Hg^2+^ in in HEPES buffer solution (10 mM, pH 7.4, containing 50% EtOH [74]. Sensor **45** showed the specific fluorescence enhancement at 664 nm (λ_ex_ = 590 nm) with large Stokes-shift after addition of Hg^2+^ with the fluorescent color change from colorless to deep red. Sensor also showed the colorimetric response with changing the color from colorless from dark blue with naked-eyes. The determined LOD for Hg^2+^ is 1.87 ppb. Mechanism of colorimetric and fluorescence response were explained by desulfurization-cyclization reaction promoted by mercury ions, resulting in the formation of spirolactum ring-opening products. Sensor **45** was also applied in different real water samples.

Asif et al. [75] reported a water soluble *p*-sulphonatocalix[4]arene derived sensor **46** appended with a rhodamine dye for the detection of Hg^2+^ in aqueous medium (Figure 15). Sensor gives spectral changes after addition of Hg^2+^ with ‘turn-on’ fluorescent response at 574 nm with the specific color change from colorless to pink. The detection limit for Hg^2+^ sensing was 3.55 × 10^−13^ mL^−1^. The sensing mechanism was explained by inhibition of PET and the fluorescence enhanced due to chelation-enhanced fluorescence (CHEF) after forming a complex **46**-Hg^2+^ in 1:1 binding stoichiometric. Zifan et al. [76] reported a sensor **47** (Figure 15) containing conjugated dyad quinolone-benzothiazole and rhodamine for the ratiometric detection of Hg^2+^ in DMF-H_2_O (7/3, *v*/*v*). Upon excitation at 390 nm, the emission band at 504 nm was decreased with the addition of Hg^2+^, while the rhodamine emission intensity at 613 nm was gradually increased due to the FRET with the sensitivity limit of 0.2 µM (Figure 16). In this process, the conjugated dyad serve as a donor and the rhodamine as an acceptor. Sensor **47** was also successfully applied in living cells. In addition, significant color change from colorless to pink attributed by Hg^2+^ induced by the opening of spirolactum ring.

Saswati et al. [77] reported a rhodamine coupled copillar[5]arene sensor **48** (Figure 17) for the selective sensing of Hg^2+^ ions in CH_3_CN medium. Sensor **48** was non-fluorescent, while showed strong fluorescence enhancement at 573 nm with the addition of Hg^2+^, and the color changed from colorless to pink due to the complextion-induced spirolactam ring opening mechanism. Sensor **48** formed a complex with Hg^2+^ in 1:1 stoichiometry, and can be applied to detect Hg^2+^ down to 28.5 nM. Jian-Peng et al. [78] reported a supramolecular sensor **49** via host–guest inclusion complexation between the host rhodamine hydrazone functionalized pillar[5]arene and the guest bis-pyridinium derivative (Figure 17). Sensor **49** showed both chromogenic and fluorogenic for the detection of Hg^2+^ in DMSO/H_2_O (6:4, *v*/*v*). The absorbance and fluorescence respective at 562 and 585 nm was enhanced with the addition of Hg^2+^ due to the complexation-induced opening of spirolactam ring. The estimated LOD with the UV–vis and fluorescence methods were 4.07 × 10^−7^ M and 1.69 × 10^−8^ M, respectively. In addition, the inductively coupled plasma data supported the ability of sensor to remove Hg^2+^.

Jiwen et al. [79] reported a rhodamine B based sensor **50** (Figure 17) for the detection of Hg^2+^. Sensor was synthesized by combining rhodamine B fluorophore with the thiophene-triazole unit as an ionophore. With the addition of Hg^2+^, sensor **50** showed significant fluorescence enhancement at 585 nm and also the absorbance increased at 560 nm that turned the colorless solution of **50** in to red. With high sensitivity, the fluorescence enhancement of **50** can be applied to detect Hg^2+^ down to 16 nM. Sagar et al. [80] reported the sensor **51** (Figure 17) containing two rhodamine units linked with 2,6-pyridinedicarboxaldehyde for the selective detection of Hg^2+^ in DMSO:H_2_O (1:1; *v*/*v*). After addition of Hg^2+^, the absorption band at 530 nm enhanced significantly with a visual color change from colorless to pink. While in emission spectra, new fluorescence band appeared at 562 nm due to the conversion of closed form of spirolactum ring of rhodamine to its ring opened form on both side of pyridine ring. The detection limit was obtained as 26 nM. Sensor **51** was applied for the real water samples for practical application.

Zixiang et al. [81] reported a rhodamine 6G based sensor **52** (Figure 18) for the selective detection of Hg^2+^ in DMSO/H_2_O (7/3, *v*/*v*). The non-fluorescent sensor showed a significant fluorescence enhancement at 581 nm after addition of Hg^2+^. In UV–vis spectra, the absorption band at 538 nm become stronger after addition of Hg^2+^ with the colorimetric response from colorless to red. The hydrogel of **52** was also prepared and applied for the reversible sensing of Hg^2+^. The sensing mechanism was described by the blocking of PET process upon complexation in 1:1 stoichiometry and the rigidity of the sensor promotes to the chelation-enhanced fluorescence (CHEF) effect. The detection limit of probe for Hg^2+^ detection is 14.9 nM. Yuesong et al. [82] have reported a novel rhodamine–naphthalene derivative **53** (Figure 18) for the Hg^2+^ detection in CH_3_CN-H_2_O (7/3, *v*/*v*). Absorption and emission peak of **53** enhanced respectively at 554 and 604 nm upon addition of Hg^2+^. Sensor **53** is non-fluorescent due to the spirolactam structure of rhodamine moiety; however, spirolactam ring was opened in presence of Hg^2+^, and give the colorimetric and fluorescent response (Figure 19). The LOD of sensor **53** for Hg^2+^ detection are 0.12 μM and 0.38 μM by using absorption and emission analysis, respectively. Sensor **53** was also applied for the test strips and biosensing applications. Guohua et al. [83] have reported a triazole-rhodamine conjugate **54** (Figure 18) for the selective detection of Hg^2+^ in DMF/H_2_O (1:1, *v*/*v*, Tris-HCl buffer, pH = 7.4). Free rhodamine sensor showed no fluorescence. However, the complexation of **54** with Hg^2+^ in 2:1 stoichiometry, the emission enhanced at 557 nm with the color changed from colorless to pink due to the spirolactam ring opening mechanism. The calculated LOD is 1.61 nM. UV–vis spectra also support the same process by exhibiting absorbance band at 563 nm. The Hg^2+^ detection by sensor **54** was applied for fluorescence imaging in HeLa cells. In another approach, Zhao et al. [84] reported a ferrocenyl containing rhodamine B based sensor **55** (Figure 18) for the detection of Hg^2+^ in H_2_O/THF (4:1, *v*/*v*). Sensor **55** showed fluorescence off-on response at 590 nm with the addition of Hg^2+^ due to the formation of a complex in 1:1 stoichiometry. Sensor **55** showed a low detection limit of 16 nM and fast response time (<3 min). Mechanism of Hg^2+^ detection is attributed by desulfurization annulation that triggers the spirolactam ring-opening. Sensor **55** was applied for monitoring of intracellular Hg^2+^ ions in living cells.

Zhong et al. [85,86] have reported three rhodamine 6G derivative **56**–**58** (Figure 20) for the fluorescent turn-on sensing of Hg^2+^ in DMSO/H_2_O (7:3, *v*/*v*). Because of the complexation-induced spirolactam ring opening, the weakly emissive **56**, **57**, and **58** showed significant fluorescence enhancement respectively at 582, 578, and 560 nm. The sensors **56**, **57**, and **58** showed the LOD of 13.4, 15.6, and 16.1 nM respectively for Hg^2+^. In addition, the absorption of **56**, **57**, and **58** enhanced respectively at 536, 537, and 534 nm leading to the naked-eye detectable color change from colorless to pink. Recently, Wei et al. [87] applied the fluorescent sensor **59** (Figure 20) encapsulated in the hydrogel microsphere for the detection of Hg^2+^ by using a microfluidic device. The non-fluorescent sensor **59** showed fluorescence enhancement with the color change to red upon addition of Hg^2+^. Sensor **59** showed a reversible response with EDTA and KI. The limit of detection for Hg^2+^ is 120 nM. Hydrogel microsphere probe was also applied for the detection of Hg^2+^ in real water samples.

Xuliang et al. [88] developed a dansyl-peptide based Hg^2+^ selective sensor (**60**, dansyl-Glu-Cys-Glu-Trp-NH_2_). Sensor **60** showed two emission maxima at 337 and 550 nm due to the tryptophan and dansyl fluorophores. Addition of Hg^2+^ caused chelation-induced fluorescence enhancement at 550 with a blue-shift to 505 nm, and the FRET from tryptophan (donor) to dansyl (acceptor) caused quenching at 337 nm. Sensor **60** can be applied to detect Hg^2+^ down to 23.0 nM. Sequentially, the in situ generated **60**-Hg^2+^ complex was applied for the sensing of biothiols. The lanthanide-complexes of Tb^3+^ are also applied for the fluorescent sensing of Hg^2+^. For example, the ratiometric sensor **61** (Figure 21) based on the lanthanide coordination polymers (CPs) between Tb^3+^, guanine monophosphate (GMP) and luminol was developed for the detection of Hg^2+^ [89]. The addition of Hg^2+^ leads to the decrease of Tb^3+^ luminescence at 548 nm due to the higher coordination between Hg^2+^ and GMP, which inhibits energy transfer from GMP to Tb^3+^. While, the fluorescence of luminol at 430 nm increased due to the aggregation-induced emission phenomenon. The ratiometric response of **61** for Hg^2+^ can be detected down to 1.3 nM. Sensor **61** was successfully used for the determination of Hg^2+^ in tap water.

Peng et al. [90] reported a dansyl based sensor **62** (Figure 21) for the fluorescence turn-on detection of Hg^2+^ in HEPES buffer. The emission of **62** at 545 nm was weak based on a single dansyl group. The formation of a complex **62**-Hg^2+^ in 2:1 stoichiometry resulted conformational adjustment that reduced the distance between two dansyl groups and formed the dansyl dimer (Figure 22). The monomer-excimer mechanism resulted significant fluorescence enhancement at 515 nm with the sensitivity limit of 22.65 nM. Muzey et al. [91] reported a naphthalimide-sulfamethizole conjugated sensor **63** (Figure 21) for the ratiometric detection of Hg^2+^ in DMSO/HEPES medium (1:99, *v*/*v*). The strong fluorescence from the monomeric form of **63** at 390 nm was quenched and a new band appeared at 483 nm due to the complexation-induced formation of excimer. Sensor **63** formed a complex with Hg^2+^ in 2:1 ratio that bring the naphthalimide close together to form the excimer. Sensor **63** showed LOD of 14.7 nM, and capable to quantify Hg^2+^ concentration in real water samples. Tapashree et al. [92] reported a pyrene-hydroxyquinoline conjugated azine based Schiff base **64** (Figure 21) for the selective detection of Hg^2+^ in ethanol-H_2_O (9:1 *v*/*v*) medium. The absorption bands of **64** at 240 and 290 nm were enhanced with the appearance of a new band at 450 nm upon addition of Hg^2+^. The color also changed from lemon yellow to golden yellow attributed to deprotonation (–OH) upon coordination with Hg^2+^ in 1:1 stoichiometry. The monomeric and excimer emission bands centered at 385 and 447 nm of **64** were enhanced due to CHEF effect that inhibited the C=N isomerization and suppressed the PET. Sensor **64** showed the LOD of 0.22 μM. The monomer-excimer based fluorescent sensor **65** (Figure 21) was reported for the selective detection of Hg^2+^ in CH_3_CN/DMSO (99:1) [93]. The pyrene appended calix[4]arene sensor **65** showed monomer emission at 395 nm. Upon complexation with Hg^2+^ in 2:1 ratio, the excimer emission enhanced at 472 nm with the sensitivity limit of 8.11 nM. Sensor **65** also showed fluorescence response towards Ag^+^. Using Hg^2+^ and Ag^+^ as two chemical inputs, the changes in the fluorescence of **65** was studied to mimic the INHIBITION and IMPLICATION logic gates.

#### 3.1.3. Reaction-Based Chemosensors

Chemodosimeter, an irreversible reaction based detection approach, where the probe contains a reactive site that interacts with the target analyte and formed a new product that emits differently from the original probe molecule. In compared to the reversible fluorescent sensors, the reaction-based fluorescent sensors showed better selectivity and specificity due to the structural changes occurred upon chemical reactions with the target analyte. The 7-hydroxycoumarin-derived carbonothioate-based sensor **66** (Figure 23) was designed and synthesized by Xiwei et al. [94] for the detection of Hg^2+^ at the maximum emission wavelength 455 nm in water medium. Upon interaction with Hg^2+^, sensor **66** showed a large fluorescence enhancement because of the strengthening of the intramolecular charge transfer (ICT) due to the Hg^2+^ directed hydrolysis of **66** to form HgS and phenol. The sensor **66** showed a LOD of 7.9 nM. Sensor **66** was successfully applied to detect Hg^2+^ in different water samples (river water), living cells and in zebrafish. With similar sensing mechanism, the multi-analytes selective chemodosimeter **67** (Figure 23) based on naphthalene fluorophore was developed for the rapid detection of Hg^2+^, hydrazine and H_2_S in C_2_H_5_OH [95]. This probe possesses multiple reactive groups phenyl thiobenzoate, carbon-carbon double bond α,β-unsaturated ketone for the detection of Hg^2+^, H_2_S and hydrazine, respectively (Figure 24). With Hg^2+^, the phenyl thiobenzoate detached from **67** and formed HgS and phenol that resulted in the fluorescence quenching at 580 nm. The Hg^2+^ sensitivity limit of **67** is 1.10 μM, and applied for the detection of Hg^2+^ in paper test strips and environmental water samples (seawater, tap water, and mineral water). In another work, the naphthalene derived probe **68** (Figure 23) was introduced for the selective chemodosimetric detection of Hg^2+^ in DMSO/H_2_O (1:3, *v*/*v*) [96]. After addition of Hg^2+^, the fluorescence intensity enhanced at 444 nm and 644 nm due to desulfurization reaction of **68** leading to the formation of HgS and phenol. The probe **68** showed a LOD of 48.79 nM for Hg^2+^, and applied successfully for quantifying Hg^2+^ in various environmental and beverages samples. In addition, probe **68** was applied for the detection of hydrazine.

Zhixiu et al. [97] synthesized a coumarin based sensor **69** (Figure 23) with the conjugation of thiourea for turn-on detection of Hg^2+^ in EtOH/H_2_O (2:8, *v*/*v*) over a broad pH range of 1–11. Addition of Hg^2+^ to the sensor **69** solution induced a hypsochromic shift of the UV–vis absorption band at 360 nm to 340 nm. Additionally, a gradual enhancement in the fluorescence emission intensity was observed at 475 nm with the detection limit of 1.46 × 10^−7^ M. The sensing mechanism of probe **69** for fluorescent detection of Hg^2+^ ion is proposed as weakly fluorescent probe **69** readily binds with Hg^2+^ ion due to the strong interaction between sulphur atom and thiophilic Hg^2+^ ion. Then, a desulfurization and cyclization process occurred to form strong fluorescence. Sensor **69** was applied in real waste water sample for detecting Hg^2+^. The coumarin-based ratiometric fluorescent probe **70** (Figure 23) was reported for the sensing of Hg^2+^ ion in ethanol and HEPES buffer (1:9, *v*/*v*) medium [98]. Probe **70** was synthesized by the catalytic reaction between coumarin-red dye and DL-dithiothreitol. Probe showed two characteristic emission band at 495 nm and 600 nm, while addition of Hg^2+^, the band at 495 nm was disappeared and the second band at 600 nm was significantly enhanced. The detection limit of probe for Hg^2+^ is 1.6 nM. The DL-dithiothreitol moiety of **70** serve as the recognition receptor for Hg^2+^. After recognition, the DL-dithiothreitol moiety detached from **70** and formed the α,β-unsaturated ketone. This chemodosimeter based probe was applied for the detection of Hg^2+^ ion in real water sample, and in living cells and zebrafish.

The fluorescent probe **71** (Figure 23) was developed by the reaction of 6-hydroxy-2-naphthaldehyde and dimethylcarbamothioic for the detection of Hg^2+^ in aqueous medium [99]. After addition of Hg^2+^, the fluorescence of **71** was enhanced at 443 nm due to the Hg^2+^-catalyzed desulfurization reaction to form HgS and 6-hydroxy-2-naphthaldehyde. The LOD of **71** for Hg^2+^ was calculated to be 39.28 nM. Probe was applied for the detection of Hg^2+^ in test paper strips and in real water samples. Probe **71** also showed selective fluorescence response for the detection of H_2_S. The chemodosimetric probe **72** (Figure 23) based on perylene diimide dye was designed for the detection of Hg^2+^ in THF-H_2_O (1:9, *v*/*v*) [100]. Highly fluorescent **72** showed a significant fluorescence quenching at 667 nm upon interaction with Hg^2+^, where the butynoxy group serve as the reactive site for Hg^2+^ detection. The LOD of 60 nM and 33 nM was determined by UV–vis and emission methods, respectively. Sensor were successfully applied detection of Hg^2+^ in blood serum and urine and bioimaging in MG-63 cells. The thioxothiazolidin-coumarin based chemodosimeter **73** (Figure 23) was introduced for the selective sensing of Hg^2+^ in HEPES-DMSO (99/1, *v*/*v*) [101]. After interaction with Hg^2+^, the emission band at 630 nm was blued-shifted and enhanced significantly at 580 nm. Remarkable changes were obtained in absorption at 530 nm with decreasing intensity and appearance of a new band at 485 nm, and also the color changed from dark to light pink. The detection limit of Hg^2+^ was estimated to be 15.6 μM and 15.1 μM by absorbance and fluorescence methods, respectively. Sensing of Hg^2+^ by sensor **73** is based on the desulfurization reaction of thiocarbonyl to carbonyl, which was supported by ^13^C NMR and HRMS-ESI analyses. Probe **73** was applied for detecting Hg^2+^ in living cells by bioimaging experiment.

The phenothiazine derived chemodosimetric probe **74** (Figure 25) containing dithioacetal unit was applied for the detection of Hg^2+^ [102]. Upon interaction with Hg^2+^, the UV–vis absorption band of **74** at 320 nm was shifted to 390 nm with the appearance of yellow color, while the weakly fluorescent **74** showed significant fluorescence enhancement with a remarkable red-shift from 455 nm to 610 nm due to intramolecular charge transfer process (ICT). With high sensitivity, the fluorescence change of **74** was applied to detect Hg^2+^ down to 21.2 nM. Sensor **74** was applied successfully to detect Hg^2+^ in drinking water and live cells. With the similar sensing mechanisms, the Hg^2+^ selective fluorescent probes **75**–**79** were reported. The AIE active chemodosimeter **75** (Figure 25) was reported for the fluorescent turn-on sensing of Hg^2+^ in PBS buffer (10 mM, pH 7.4, containing 1% DMSO). Sensor **75** showed a weak emission at 475 nm. With the Hg^2+^ directed hydrolysis of **75** at the dithioacetal unit to aldehyde enhanced the ICT effect and also restricted the intramolecular rotations that amplified the fluorescence at 495 nm based on AIE effect. The detection limit of sensor **75** for Hg^2+^ is 36 nM. Sensor **75** was applied for the detection of Hg^2+^ in real water sample and in living cell imaging [103].

Zhonglong et al. [104] reported a camphor based fluorescent turn-on probe **76** (Figure 25) for the detection of Hg^2+^ with large Stokes shift of 153 nm in 99% PBS buffer medium. Probe **76** exhibited a maximum absorption at 322 nm that reduced remarkably while a new band emerged at about 355 nm after adding Hg^2+^. With high sensitivity, the fluorescence of **76** was enhanced at 518 nm upon interaction with Hg^2+^. The 1,3-dithiane unit of the **76** can be deprotected into a formyl group under the function of Hg^2+^, thus the probe **76** is transformed into compound **77**-CHO. The detection limit of **76** is 19.3 nM for Hg^2+^. Sensor was applied in cell imaging experiment and to quantify Hg^2+^ in environmental water samples (tap, distilled, and lake water). The bithiophene-based sensor **77** (Figure 25) was reported for the ultra-rapid detection of Hg^2+^ in aqueous medium with the fluorescent color changed from colorless to blue under UV light irradiation [105]. Significant enhancement in emission of **77** at 470 nm was observed due to the Hg^2+^ induced desulfurization reaction that strengthened the ICT upon conversion of bithiophene moiety in to aldehyde group (Figure 26). With **77**, the Hg^2+^ concentration can be detected down to 19 nM. Sensor **77** also showed selective changes in the absorption by quenching of absorbance at 334 nm and appearance of a new band at 370 nm after the addition of Hg^2+^. The fluorescence changes of **77** was applied to quantify Hg^2+^ detection in water, seafood as well as human urine samples. In addition, Sensor **77** was applied for Hg^2+^ detection by developing test paper strips and performing bio-imaging in HeLa cells.

Meiju et al. [106] reported a fluorescent sensor **78** (Figure 25) based on the naphthalimide derivative for the detection of Hg^2+^ in PBS buffer. In UV–vis absorption, the sensor **78** showed blue-shift from 461 to 417 nm with Hg^2+^. The fluorescence emission enhanced at 510 nm with increasing concentration of Hg^2+^. In the presence of Hg^2+^, the sensor **78** reacted specifically with the mercury ion to produce an aldehyde and emitted strong fluorescence, and the yellow color of the solution turned to light green. The detection limit for Hg^2+^ was found to be 40 nM. The sensor **78** was successfully applied to the living cell imaging to detect Hg^2+^ in PC-12 cells. The tetraphenylethylene (TPE) derivative **79** (Figure 25) reported by Long et al. showed aggregation-induced emission features in THF/water mixtures. TPE derivatives maintain AIE activities after grafting on fibers, however, the strong fluorescence emission at 477 nm was gradually weakened after Hg^2+^ addition due to Hg^2+^-initiated cleavage of dithioacetal moieties. The LOD reached as low as 20 nM Hg^2+^. In addition, the electrospun fibrous strips with grafted TPE and dithioacetal moieties are designed for the detection of trace Hg^2+^ with the visual change of color strip from green to blue [107].

Abani et al. [108] reported a trinuclear Zn(II)/Cd(II) Schiff base complexes **80** and **81** for the detection of Hg^2+^ in aqueous medium via chemodosimetric approach (Figure 27). Two absorption maxima at 388 nm and 390 nm were red-shifted and the colorless solution turned to distinct yellow. Complexes showed emission maxima at λ_em_ ∼ 461 nm (**80**) and 464 nm (**81**) were red-shifted to 475 and 472 nm, respectively and undergo considerable decrease in fluorescence intensity. The LOD estimated for complexes **80** and **81** to detect Hg^2+^ were 1.11 and 1.89 μM, respectively. The most probable chemodosimetric mechanism explained via the cleavage of the imine bond through hydrolysis. Results were confirmed by different spectroscopic techniques including ^1^H NMR titration.

Chunqing et al. [109] reported the monomeric BODIPY based Schiff bases **82** and **83**, and the polymeric derivative **84** for the fluorescent turn-on sensing of Hg^2+^ and Fe^3+^ in DMF/H_2_O (1:1, *v*/*v*) (Figure 27). Both the selective metal ions hydrolysed the imine linkage and formed the original BODIPY aldehyde. The emission and absorption peak of the probes were blue-shifted and enhanced. The emission of **82**, **83**, and **84** at 549, 550, and 559 nm was blue-shifted and enhanced respectively at 523, 529, and 528 nm. The probes **82**, **83**, and **84** showed the LOD of 0.21, 0.63, and 0.19 µM respectively for Hg^2+^. The polymeric probe **84** showed high sensitivity than the probes **82** and **83**. Similarly, the absorption of **82**, **83**, and **84** at 520, 545, and 548 nm was blue-shifted and enhanced respectively at 490, 499, and 501 nm.

### 3.2. Colorimetric Sensors

The colorimetric sensors provide naked-eyes detectable color change for the cost-effective of target analytes (Table 3). The ruthenium derived complex **85** (Figure 28) was reported for the colorimetric detection of Hg^2+^, where the coordination of mode of Ru^2+^ to the C-atom is changed to S-atom with the addition of Hg^2+^ (Figure 29). The Hg^2+^-prompted switch in coordination mode in **85** caused a color from dark red to light yellow, and the absorption band at 506 and 730 nm are gradually quenched with the sensitivity limit of 21 nM [110]. With the similar approach, the ruthenium complex based sensor **86** (Figure 28) was reported for the selective colorimetric detection of Hg^2+^. The color of **86** turned from red to yellow, due to the formation of a new low energy band at 410 nm, while pre-existed band at 503 declined with the addition of Hg^2+^. The detection limit of **86** for Hg^2+^ is 0.053 μM. Job’s plot confirmed the binding stoichiometry of sensor **86**-Hg^2+^ complex is 1:1 mode. For the practical applicability, the sensor was grafted into a polymer membrane and applied for the colorimetric detection of Hg^2+^ [111].

The *p*-toluenesulfonate salt of merocyanine dye **87** (Figure 28) was reported for the colorimetric sensing of Hg^2+^ in HEPES buffer. Sensor **87** showed two absorption band at 390 nm and 530 nm. After complexation with Hg^2+^, both the absorption bands showed decrease in the intensity with the color change from pink to colorless easily detected by naked eyes. Sensor showed a micromolar detection limit of 0.27 μM and was also applied for the visual detection of Hg^2+^ by using paper test strip of **87 [112]**. The pyrazole-based colorimetric sensor **88** (Figure 28) was developed for the detection of Hg^2+^ in semi-aqueous medium. The absorption band of **88** at 447 nm was red-shifted to 519 nm with a noticeable color change from yellow to pink upon complexation with Hg^2+^ in 2:1 stoichiometry. Sensor **88** showed LOD of 4.73 × 10^−7^ M, and was applied successfully to quantify Hg^2+^ in various environmental samples [113]. In another work, the azo dye based chromogenic sensor **89** (Figure 28) was studied for selective detection of Hg^2+^ in DMSO-H_2_O (4:1 *v*/*v*) medium. Sensor **89** showed absorption at 502 nm due to strong intermolecular charge transfer (ICT) transition with the solution color as reddish-pink. Addition of Hg^2+^ decreased the absorbance at 502 nm and blued-shifted to 395 nm. The complex formation between sensor **89** and Hg^2+^ restricted the ICT that caused blue-shift in absorbance, and the solution become colorless. Sensor **89** formed complex with Hg^2+^ in 1:1 stoichiometric through the salicylaldehyde unit and the complexation reversed with the addition of F^-^ [114].

The conjugated Schiff base receptor **90** (Figure 30) showed colorimetric response for the detection of Hg^2+^ in buffer/DMF (98:2) [115]. Only Hg^2+^ showed spectral and color changes from pale yellow to orange. The absorption band at 400 nm was increased with decreasing of **90** band at 350 nm due to the formation a complex **90**-Hg^2+^ in 2:1 stoichiometry. The limit of detection of probe for Hg^2+^ is 0.11 µM, and applied to quantify Hg^2+^ in real water samples.

Thirumalai et al. [116] reported two solid templates, mesoporous silica monoliths (MSMs) and mesoporous polymer monoliths (MPMs), immobilized with the amphiphilic chromo-ionophoric **91** (Figure 30) to develop solid-state sensors for naked-eye colorimetric sensing of Hg^2+^. Upon interaction with Hg^2+^, the solid-state sensors showed color transition from light orange to deep red due to the metal to ligand charge transfer (MLCT). The LOD for **91**-MPM and **91**-MSM sensors was estimated as 0.100 and 0.180 μg/L, respectively. Both the sensors applied successfully in real sample analysis to quantify Hg^2+^ concentrations. The solid-state sensor **92** (Figure 30) based on rhodamine B hydrazide derivative mobilized in mesoporous silica monolith was developed for the ultra-trace colorimetric detection of Hg^2+^ from aqueous medium [117]. The absorption of **92** was changed at 567 nm after each addition of Hg^2+^, and the light pink color turned to deep violet. Sensor **92** can be used to detect Hg^2+^ down to 0.61 µg/L, and applied successfully for quantifying Hg^2+^ ion in real water samples (ground, lake, and river water).

Hyokyung et al. [118] introduced a Pt complex **93** (Figure 30) coordinated with the ligands 1,2-bis[bis(pentafluorophenyl)phosphino]ethane) and 1,3-dithiole-2-thione-4,5-dithiolate for the selective colorimetric detection of Hg^2+^ in CH_3_CN/H_2_O (1:1, *v*/*v*). Complex **93** showed Hg^2+^ selective color change from yellow to vivid red due to the interaction of Hg^2+^ at >C=S, and also the absorbance at 448 nm of **93** was red-shifted to 523 nm. The colorimetric sensor **94** (Figure 30) based on the anthracene moiety was developed for the detection of Hg^2+^ in HEPES buffered CH_3_OH:H_2_O (7:3) medium. Sensor **94** showed absorbance at 414 nm was disappeared with the formation of a new red-shifted band at 498 nm after the addition of Hg^2+^. The complexation-induced spectral change caused due to ICT also showed naked-eye detectable color change from yellow to pink. With sensor **94**, the concentration of Hg^2+^ can be detected down to 220 nM. Sensor **94** was applied for the various practical applications including naked-eyes detection of Hg^2+^ using paper strips and solid silica gel, and also to quantify Hg^2+^ concentration in real water samples [119].

The benzopyran based colorimetric sensor **95** (Figure 31) was reported for the visual detection of Hg^2+^ ion in CH_3_CN/H_2_O medium (1:1, *v*/*v*). In this sensor, the dicyanomethylene-4H-chromene serve as a fluorophore whereas the dithiadioxa-monoaza crown ether as the recognition unit. Sensor **95** absorption at 517 nm showed hypsochromic shift with Hg^2+^ and a new band generated at 415 nm. Color of probe is also changed from rose red to yellow detected by naked eyes. Color and spectral changes mainly attributed by the blocking of ICT process. In fluorescence study, the strong emission peak at 645 nm was significantly quenched with Hg^2+^ due to the complex formation of **95**-Hg^2+^ in 1:1 ratio. The detection limit is 0.14 μM. For practical applications, sensor was applied for Hg^2+^ detection in real aqueous samples and live cell imaging [120].

Zhang et al. [121] reported chemosensor **96** (Figure 31) based on azobenzene for the selective detection of Hg^2+^ in HEPES buffered solution. In UV–vis study, the sensor **96** showed characteristic absorbance at 358 and 247 nm respectively due to π–π* and n–π* transitions. After interaction with Hg^2+^, the π–π* band was suppressed and blue-shifted by 13 nm. Absorption changes arising from the pull–push effect between electron-withdrawing and electron-donating groups of the azobenzene chromophore. The nitrobenzoxadiazole-antipyrine conjugate **97** (Figure 31) was studied for the colorimetric sensing of Hg^2+^ and CN^-^ in CH_3_OH: H_2_O (1: 1, *v*/*v*). Addition of Hg^2+^ evolved new absorption band at 530 nm and band at 465 was red-shifted to 485 nm leading to the color change from pale yellow to pink due to the formation of a charge-transfer complex between probe and Hg^2+^ in 2:1 stoichiometry. Sensor showed reversibility with Na_2_S and the LOD of 2.57 × 10^−8^ M Hg^2+^ [122]. The colorimetric sensor **98** (Figure 31) was reported for the detection of Hg^2+^ and Cu^2+^ in MeCN-H_2_O (1:1, *v*/*v*). The colorless solution of **98** turned brick-red to the naked eye with the addition of Hg^2+^ and Cu^2+^, respectively due to the complexation-induced LMCT. The complexation led to the formation of a new absorption at 470 nm and the quenching of sensor band at 380 nm. The estimated detection limit for Hg^2+^ is 0.95 nM, and the sensor applied for real samples analyses [123].

The azo-phenylthiourea based receptor **99** (Figure 31) was applied as a colorimetric sensor for Hg^2+^ in DMSO/H_2_O (2:1, *v*/*v*). Sensor **99** showed an absorption band centered at 365 nm, attributed to ICT of azo skeleton. Interaction with Hg^2+^ in 1:1 stoichiometry, sensor **99** generated a new ICT band at 280 nm with the significant hypsochromic shift of 85 nm. Sensor **99** showed the lower detection limit of 4.89 μM for Hg^2+^, and applied for the monitoring of Hg^2+^ in the real samples [124]. In another work, Gargi et al. reported an azo dye based colorimetric sensor **100** (Figure 31) for the detection of Hg^2+^ in 9:1 (*v*/*v*) aqueous CH_3_CN. After addition of Hg^2+^ to the colorless solution of **100** showed a new absorption band at 610 nm that enhance π-delocalization and reduced the energy of π → π* transition leading to the appearance of greenish-blue color. Job’s plot confirmed 1:1 stoichiometry between **100** and Hg^2+^. The limit of detection was 8.5 μM, and the sensor was applied for the real water analysis [125].

The ninhydrin–thiosemicarbazone based sensor **101** (Figure 31) was developed for the colorimetric sensing of Hg^2+^ in aqueous medium [126]. After complexing with Hg^2+^ in 1:1 ratio, the absorption band of **101** at 335 nm was shifted to 305 nm without any pH effect, and the solution color turned from yellow to colorless. Addition of strong chelating agent ETDA reversed the color change occurred due to the **101**-Hg^2+^ complex formation in solution. The LOD of **101** for Hg^2+^ detection was 1 μM. Gurjaspreet et al. [127] prepared an antipyrine based sensor **102** (Figure 31) for the detection of Hg^2+^ and Fe^3+^ in DMSO/H_2_O (8/2 *v*/*v*) by UV–vis method. After interaction with Hg^2+^, the absorption band of sensor at 290 nm was blue-shifted with the evolution of two bands at 255 nm and 292 nm due to the participation of azomethine linkage in the formation of Hg-N bond. The **102** LOD to detect Hg^2+^ was estimated as 0.10 mM.

## 4. Conclusions

In this review, we have summarized 102 chromo-fluorogenic chemosensors reported in the year 2020 for the sensing of mercuric ion. Most of the developed sensors are easy-to-prepare, low cost, and showed high selectivity and rapid response. In compared to colorimetric sensors, more focus is given on the development of fluorescent sensors because of the high sensitivity and their utility in monitoring intracellular Hg^2+^ ions in live cells. The majority of the summarized sensors are based on the well-known sensing mechanisms like PET, ICT, ESIPT, AIE, FRET, and excimer-monomer. The majority of the fluorescent sensors are either turn-off or turn-on, and there is need of more research on the designing of ratiometric sensors for Hg^2+^. Also, there is need of more attention in the designing of sensors applicable in pure aqueous medium over a wide pH range. Despite high sensitivity, the commercialization of sensors for real-world samples detection required great efforts on improving the sensor performance and also on fabrication methods. Therefore, future research may be focused on integrating the fascinating color change shown by the sensors even at low concentration with smartphone and other portable devices for the on-site, real-time and cost-effective detection of Hg^2+^. The paper chips, polymeric or other testing strips of sensors may be developed for the detection of Hg^2+^. The sensing mechanisms should be properly investigated to provide appropriate future directions to optimize the structure and performance for the designing of sensors. The concepts from nano and supramolecular chemistry may also be incorporated in the designing of novel sensors with improve sensing performance and to minimize the interference from other analytes in complex biological samples. We believe this review will provide new directions for designing novel and cost-effective sensors for Hg^2+^ with improved aqueous solubility, selectivity, and sensitivity.

## Figures and Tables

**Figure 1 pharmaceuticals-14-00123-f001:**
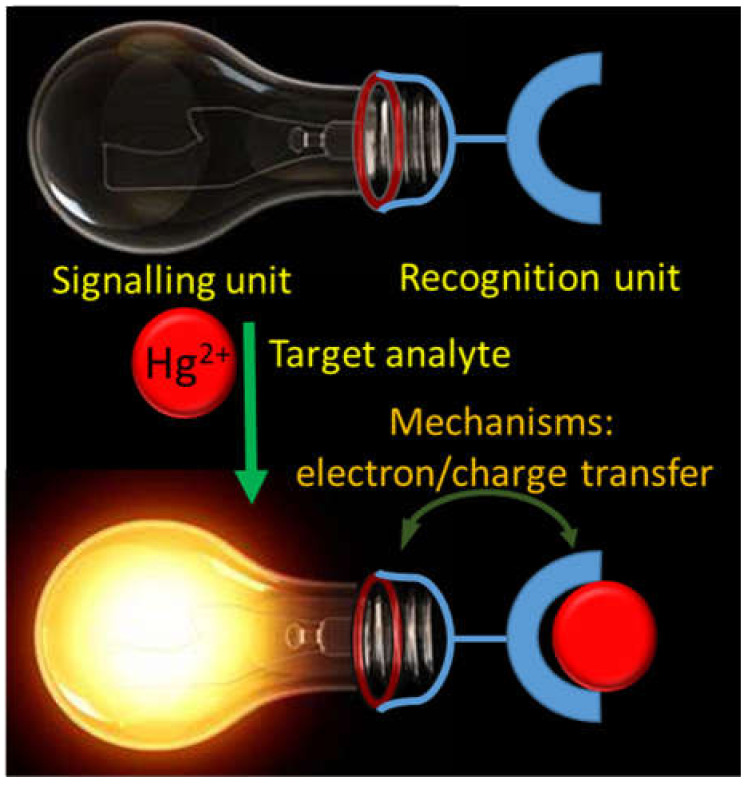
Schematic representation showing the design of a chemosensor.

**Figure 2 pharmaceuticals-14-00123-f002:**
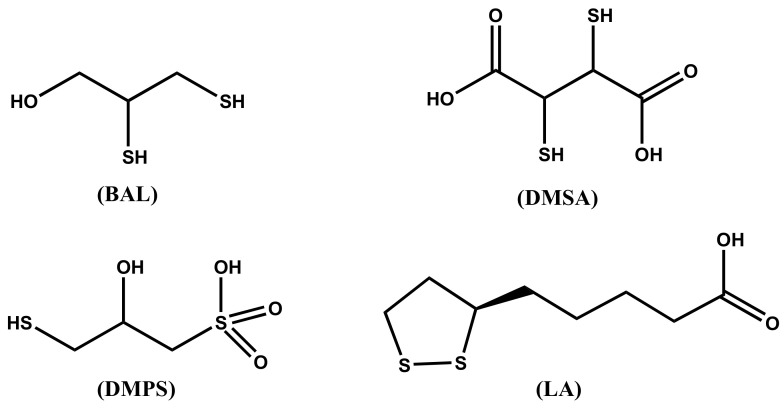
Molecular structure of the chelating agents used for mercury intoxication.

**Figure 3 pharmaceuticals-14-00123-f003:**
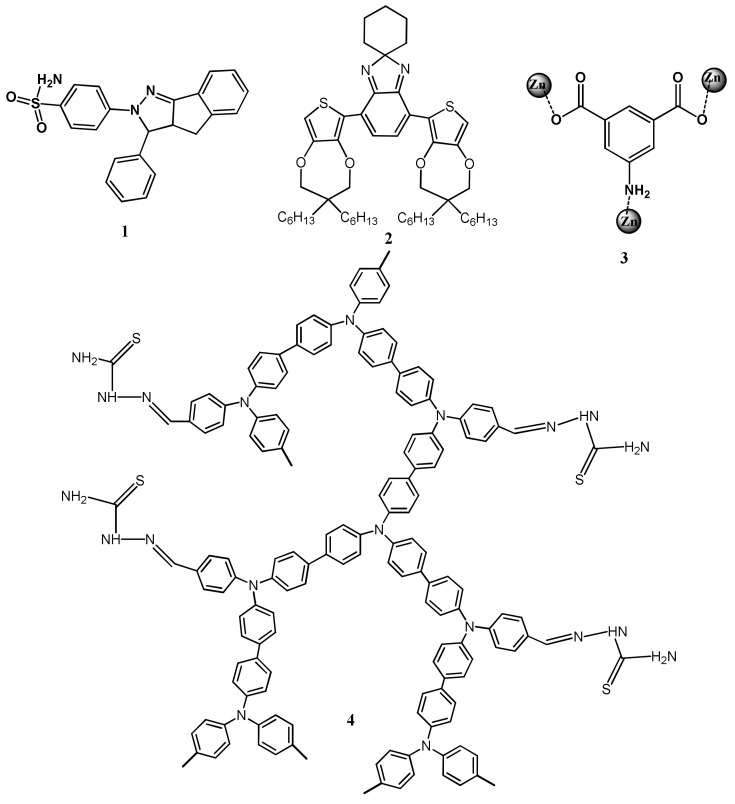
Molecular structures of the sensors **1** to **4**.

**Figure 4 pharmaceuticals-14-00123-f004:**
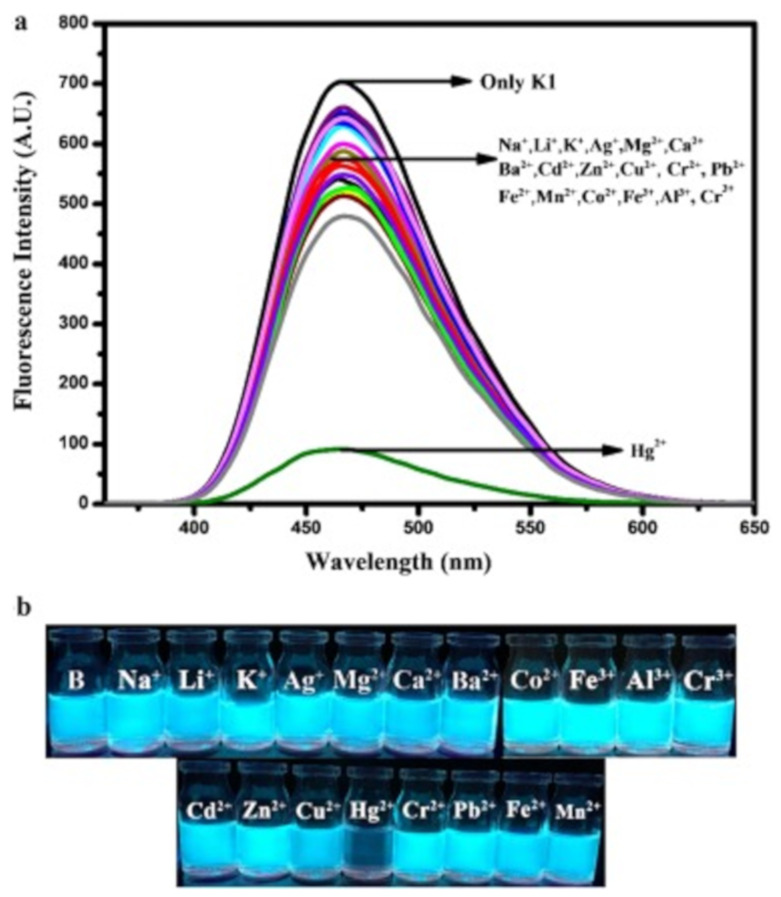
(**a**) Fluorescence spectra of 1 (K_1_ = 1) and (**b**) photographs of 1 (B = 1) under UV in the absence and presence of 20 µM metal ions in water. Reproduced with permission from [32]. Copyright 2020 Elsevier.

**Figure 5 pharmaceuticals-14-00123-f005:**
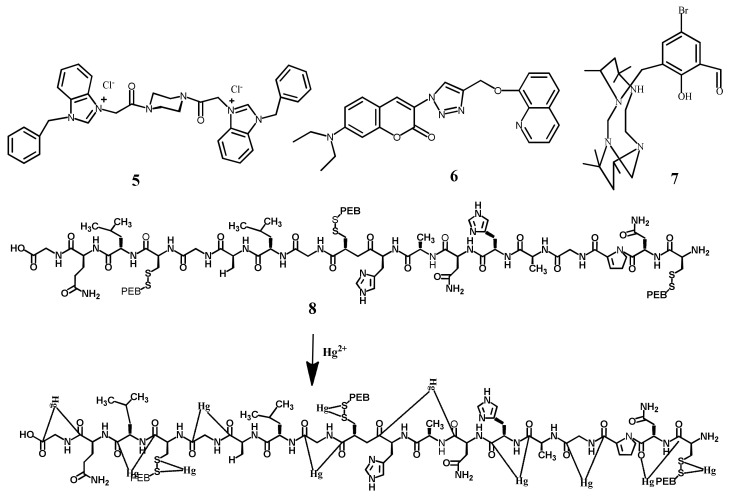
Molecular structures of the sensors **5** to **8**.

**Figure 6 pharmaceuticals-14-00123-f006:**
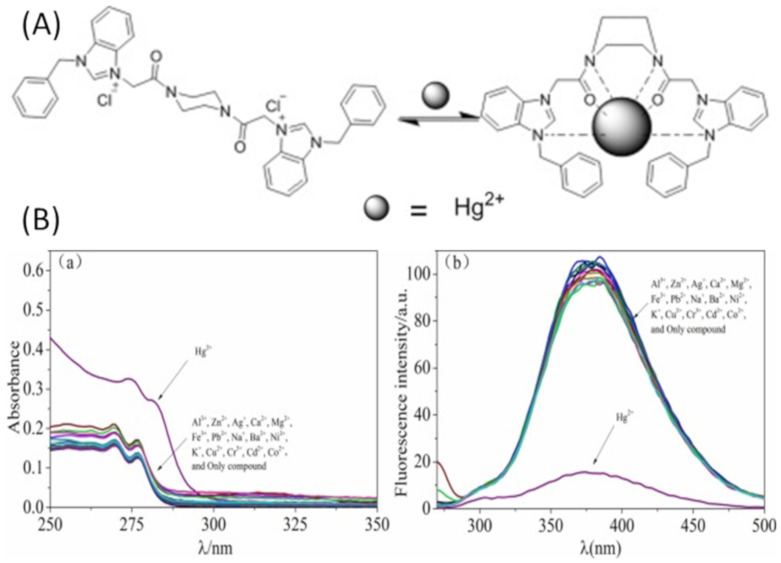
(**A**) Coordination mode of **5** with Hg^2+^, and (**B**) the UV absorption (**a**) and fluorescence (**b**) spectra of sensor 5 (10 μM) in CH_3_CN/H_2_O (1:1, *v*/*v*, pH = 7.4) upon the addition of different cations. Reproduced with permission from [36]. Copyright 2020 Elsevier.

**Figure 7 pharmaceuticals-14-00123-f007:**
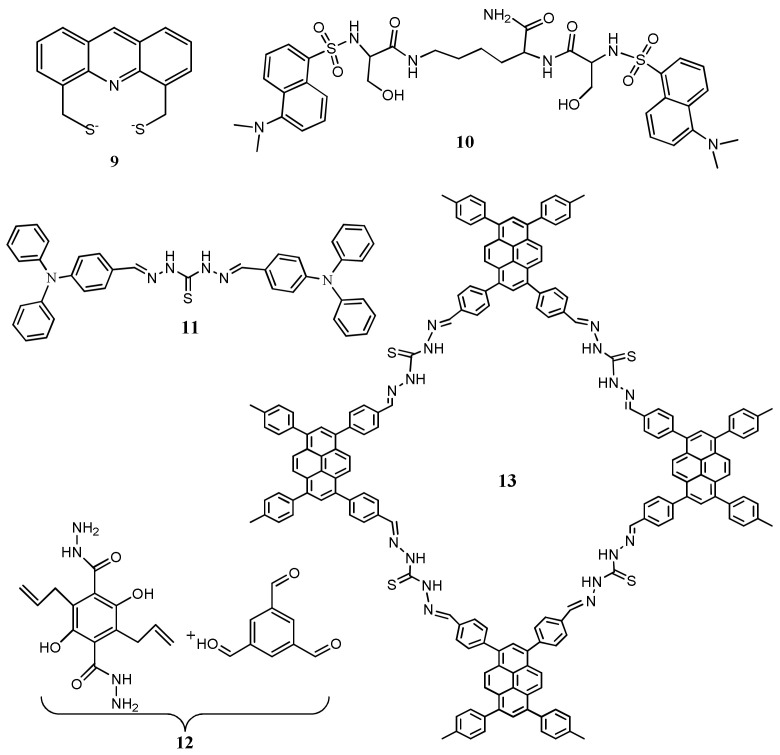
Molecular structures of the sensors **9** to **13**.

**Figure 8 pharmaceuticals-14-00123-f008:**
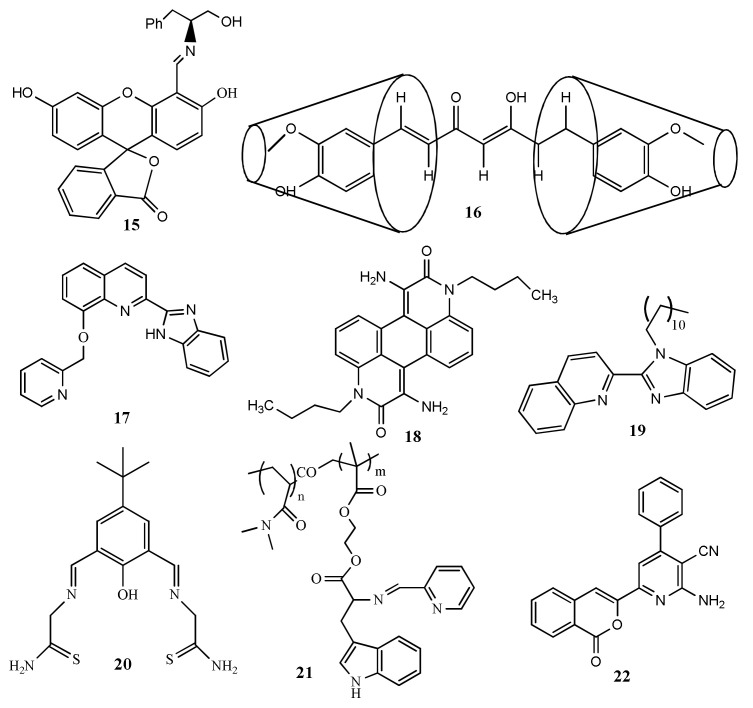
Molecular structures of the sensors **15** to **22**.

**Figure 9 pharmaceuticals-14-00123-f009:**
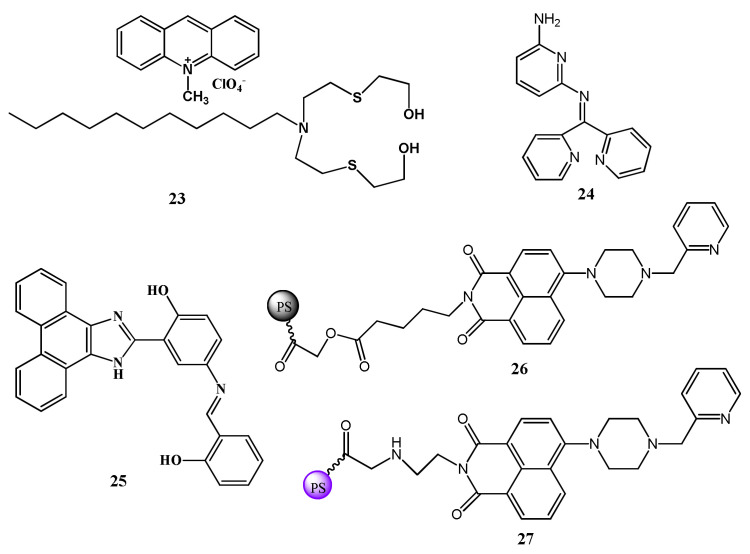
Molecular structures of the sensors **23** to **27**.

**Figure 10 pharmaceuticals-14-00123-f010:**
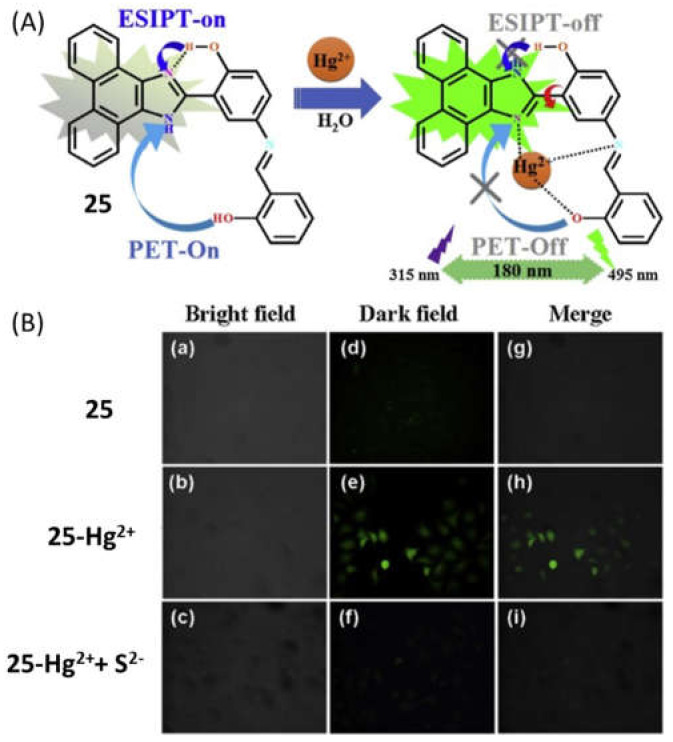
(**A**) Proposed binding mode and responding mechanism of ***25*** with Hg^2+^. (**B**) Confocal fluorescence images of HeLa cells incubated by ***25*** (**a**, **d**, and **g**), ***25***-Hg^2+^ (**b**, **e**, and **h**) and ***25***-Hg^2+^ + S^2−^ (**c**, **f**, and **i**), respectively. Reproduced with permission from [56]. Copyright 2020 Elsevier.

**Figure 11 pharmaceuticals-14-00123-f011:**
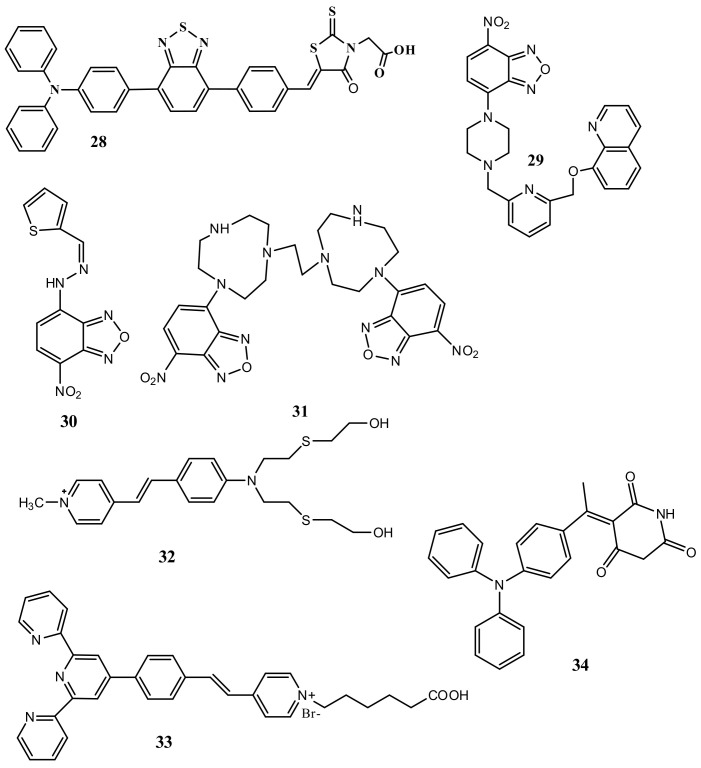
Molecular structures of the sensors **28** to **34**.

**Figure 12 pharmaceuticals-14-00123-f012:**
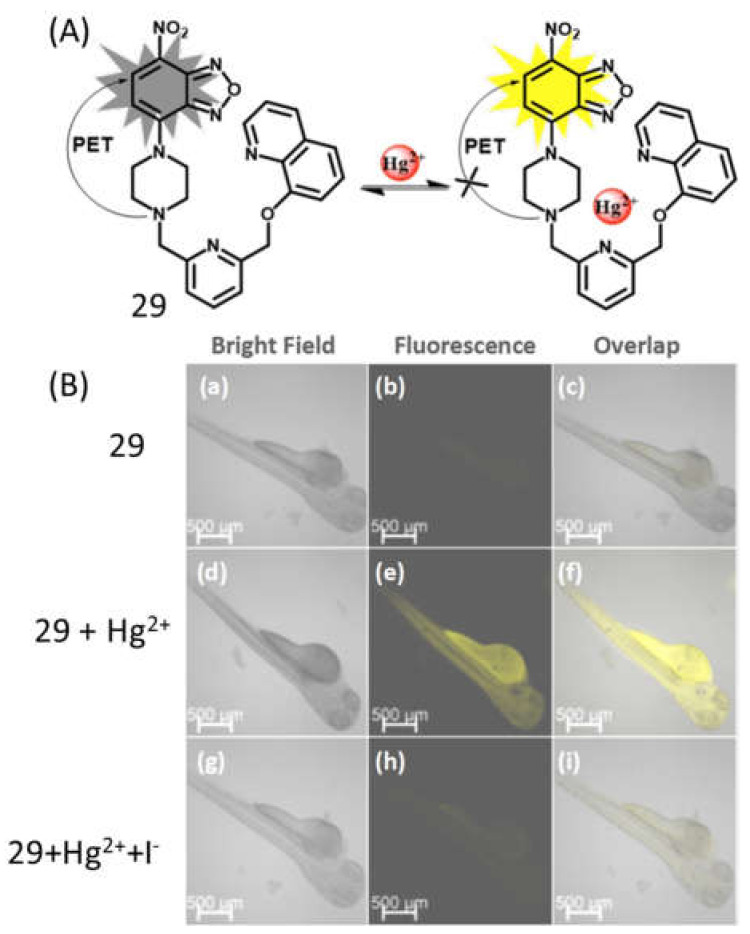
(**A**) Proposed binding mode and responding mechanism of ***29*** with Hg^2+^. (**B**) Fluorescence images of three-day-old zebrafish incubated with **29** (20 μM) at 28 °C for 30 min (**a**–**c**), subsequent treatment with Hg^2+^ (40 μM) for 30 min (**d**–**f**) and further incubation with I^−^ (40 μM) for 30 min (**g**–**i**). λ_ex_ = 488 nm; yellow channel: λ_em_ = 500–600 nm; scale bar: 500 μm. Reproduced with permission from [59]. Copyright 2020 Elsevier.

**Figure 13 pharmaceuticals-14-00123-f013:**
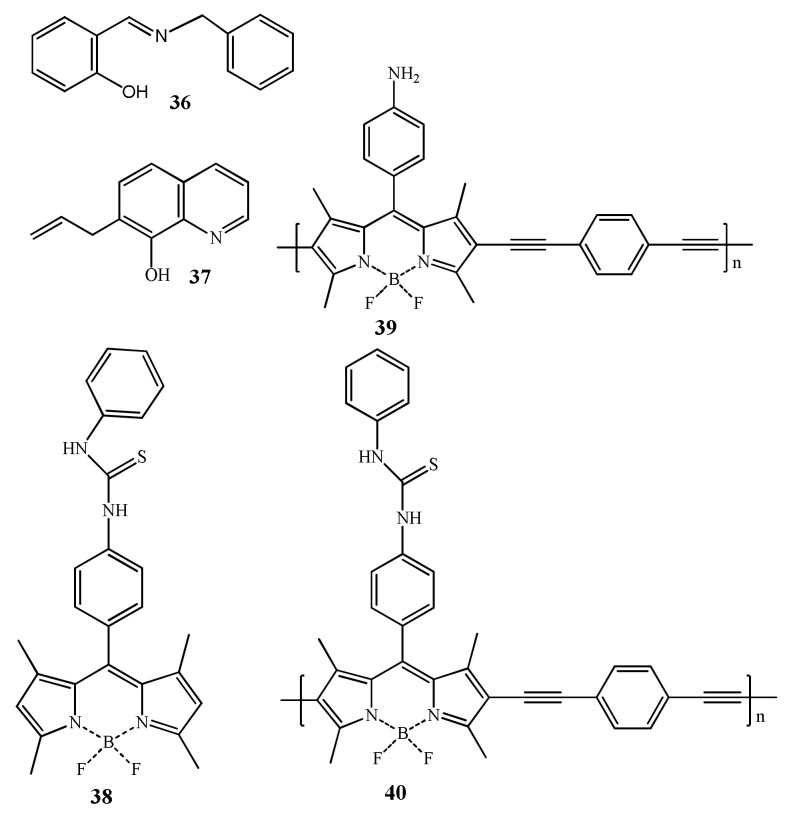
Molecular structures of the sensors **36** to **40**.

**Figure 14 pharmaceuticals-14-00123-f014:**
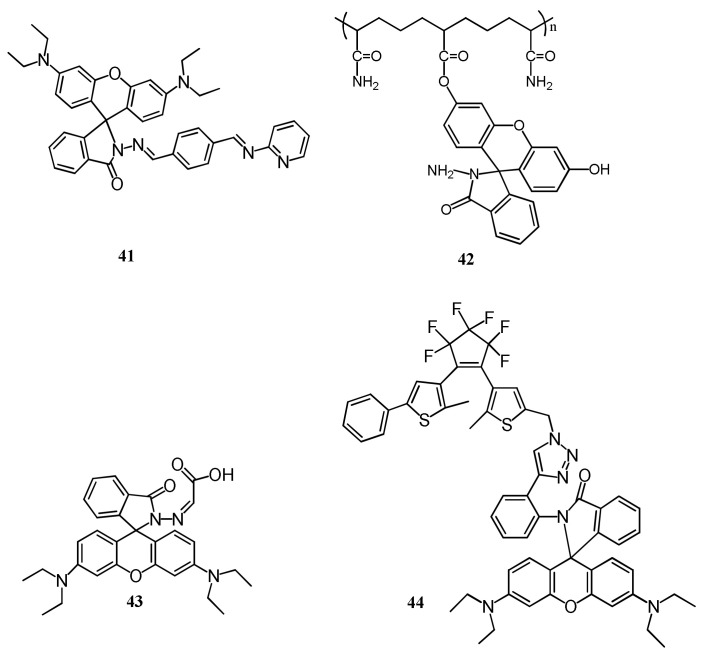
Molecular structures of the sensors **41** to **44**.

**Figure 15 pharmaceuticals-14-00123-f015:**
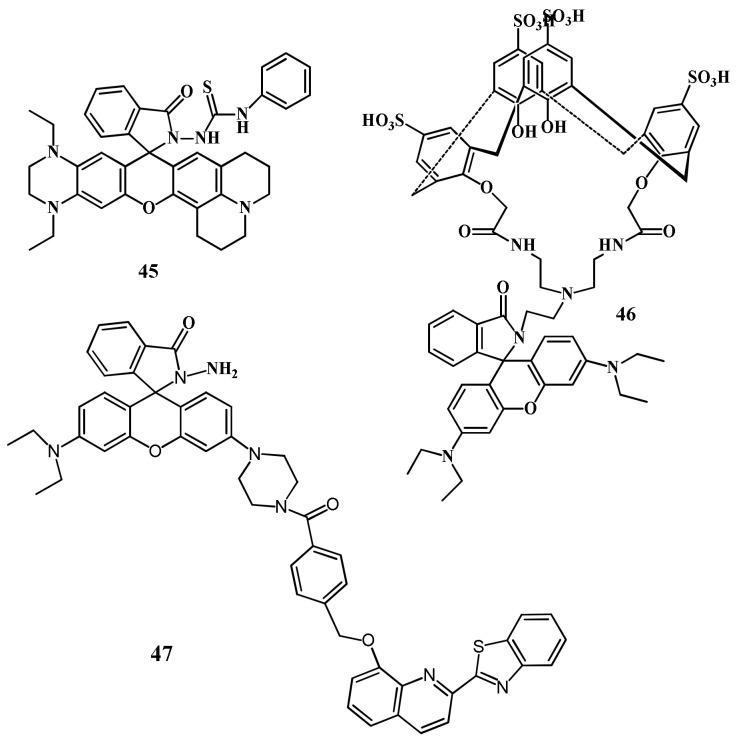
Molecular structures of the sensors **45** to **47**.

**Figure 16 pharmaceuticals-14-00123-f016:**
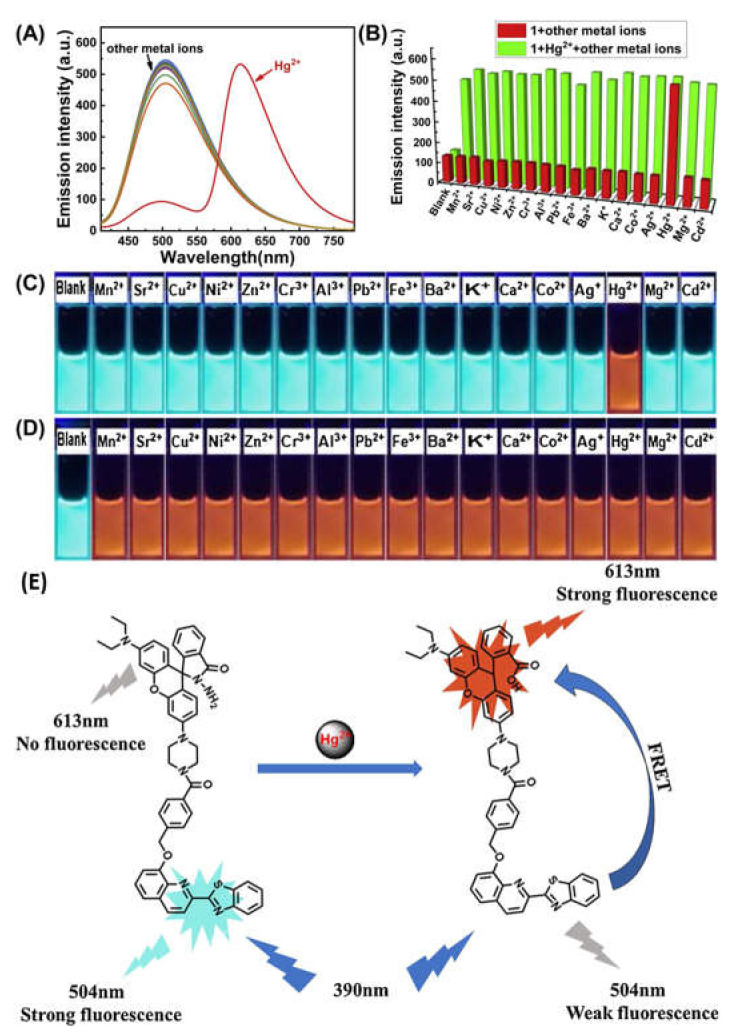
(**A**) Fluorescence responses of 47 (1 in image represents sensor 47) (10 μM) in the presence of various metal ions; (**B**) Fluorescence intensity at 613 nm of 47 (red bars: 47 with other metal ions, green bars: 47 with other metal ions and Hg^2+^, *λ*_ex_ = 390 nm). (**C**) The color image of 47 (10 μM) with other metal ions. (**D**) The fluorescence photo of 47 (10 μM) upon addition Hg^2+^ (20 eq.) in the existence of other metal ions (20 eq.). (**E**) The FRET process of 47 detecting Hg^2+^. Reproduced with permission from [76]. Copyright 2020 Elsevier.

**Figure 17 pharmaceuticals-14-00123-f017:**
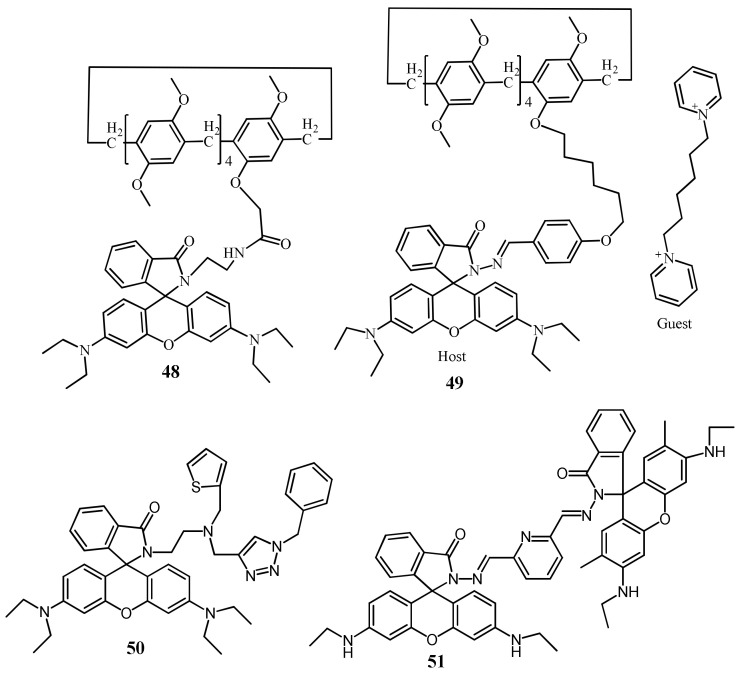
Molecular structures of the sensors **48** to **51**.

**Figure 18 pharmaceuticals-14-00123-f018:**
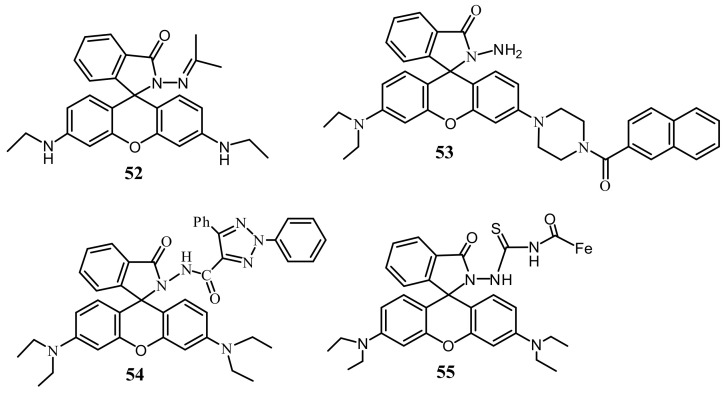
Molecular structures of the sensors **52** to **55**.

**Figure 19 pharmaceuticals-14-00123-f019:**
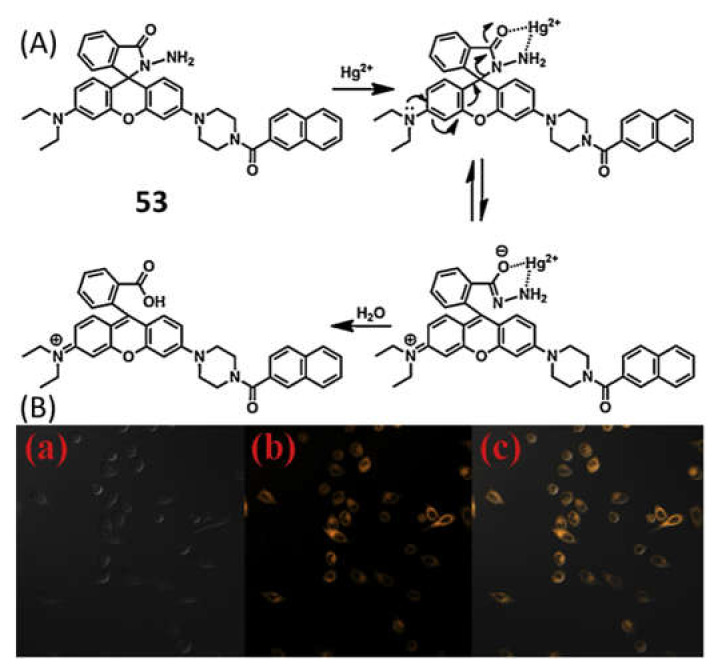
(**A**) Sensing mechanism of **53** for Hg^2+^. (**B**) Confocal fluorescence images of HeLa cells with an excitation filter of 488 nm. Probe **53** loaded HeLa cells: (**a**) Bright field image; (**b**) Dark field image; (**c**) Merged images of **a** and **b**. Reproduced with permission from [82]. Copyright 2020 Elsevier.

**Figure 20 pharmaceuticals-14-00123-f020:**
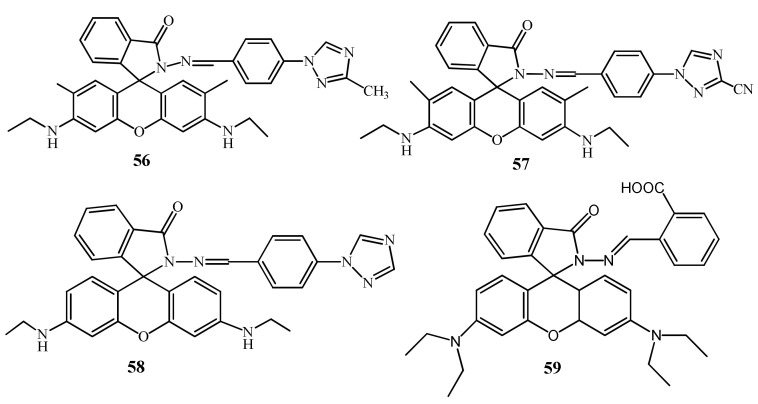
Molecular structures of the sensors **56** to **59**.

**Figure 21 pharmaceuticals-14-00123-f021:**
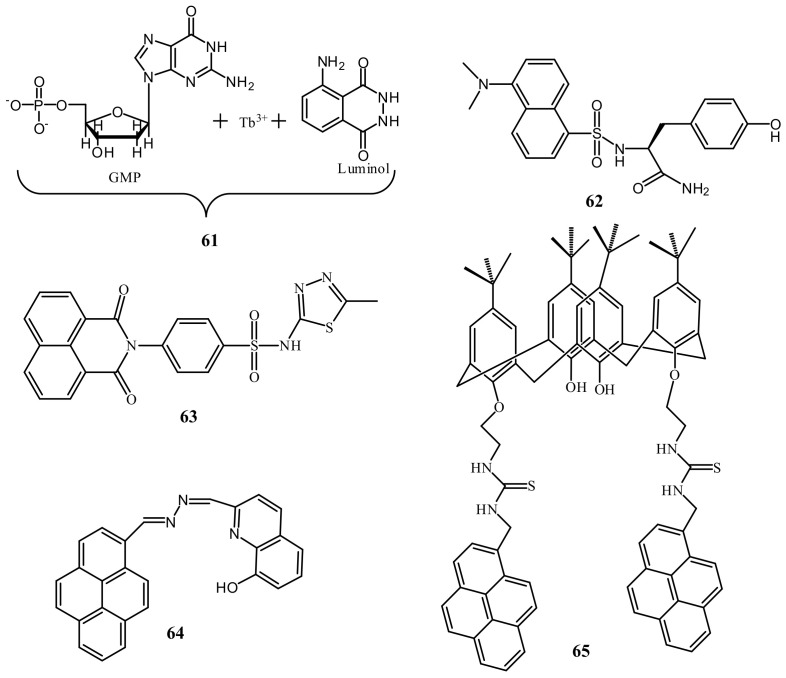
Molecular structures of the sensors **61** to **65**.

**Figure 22 pharmaceuticals-14-00123-f022:**
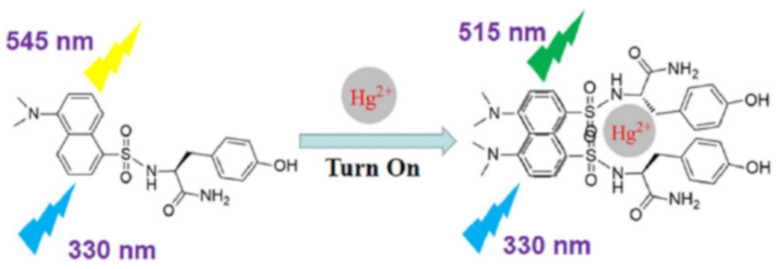
Proposed principle of **62** for detecting Hg^2+^. Reproduced with permission from [90]. Copyright 2020 Elsevier.

**Figure 23 pharmaceuticals-14-00123-f023:**
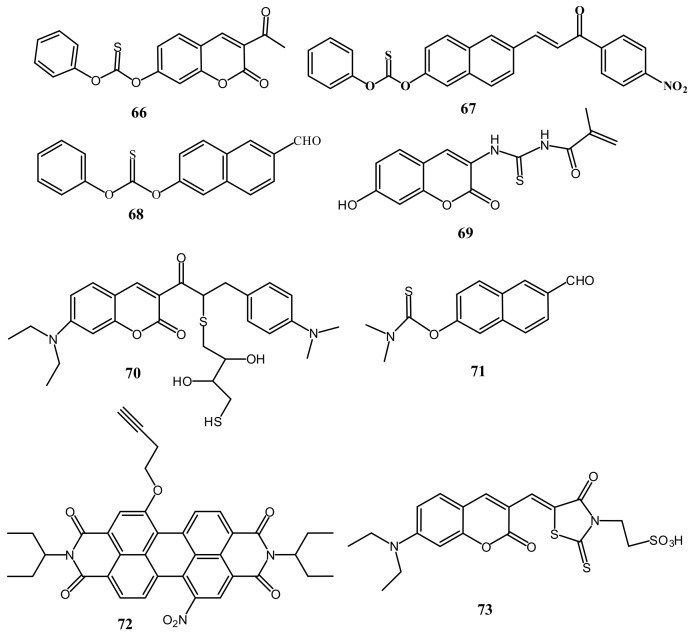
Molecular structures of the sensors **66** to **73**.

**Figure 24 pharmaceuticals-14-00123-f024:**
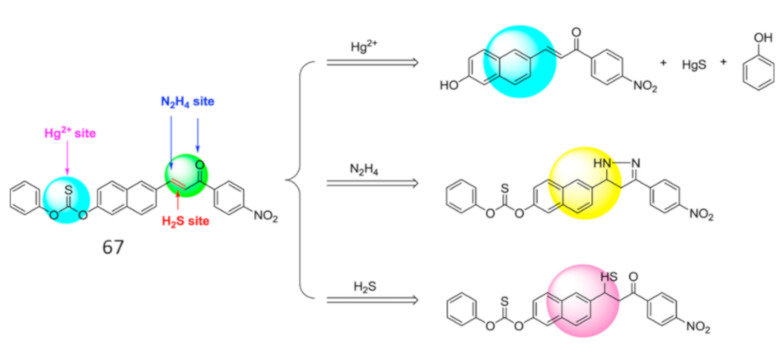
Reaction-based sensing mechanism of 67 with Hg^2+^, N_2_H_4_ and H_2_S. Reproduced with permission from [95]. Copyright 2020 Elsevier.

**Figure 25 pharmaceuticals-14-00123-f025:**
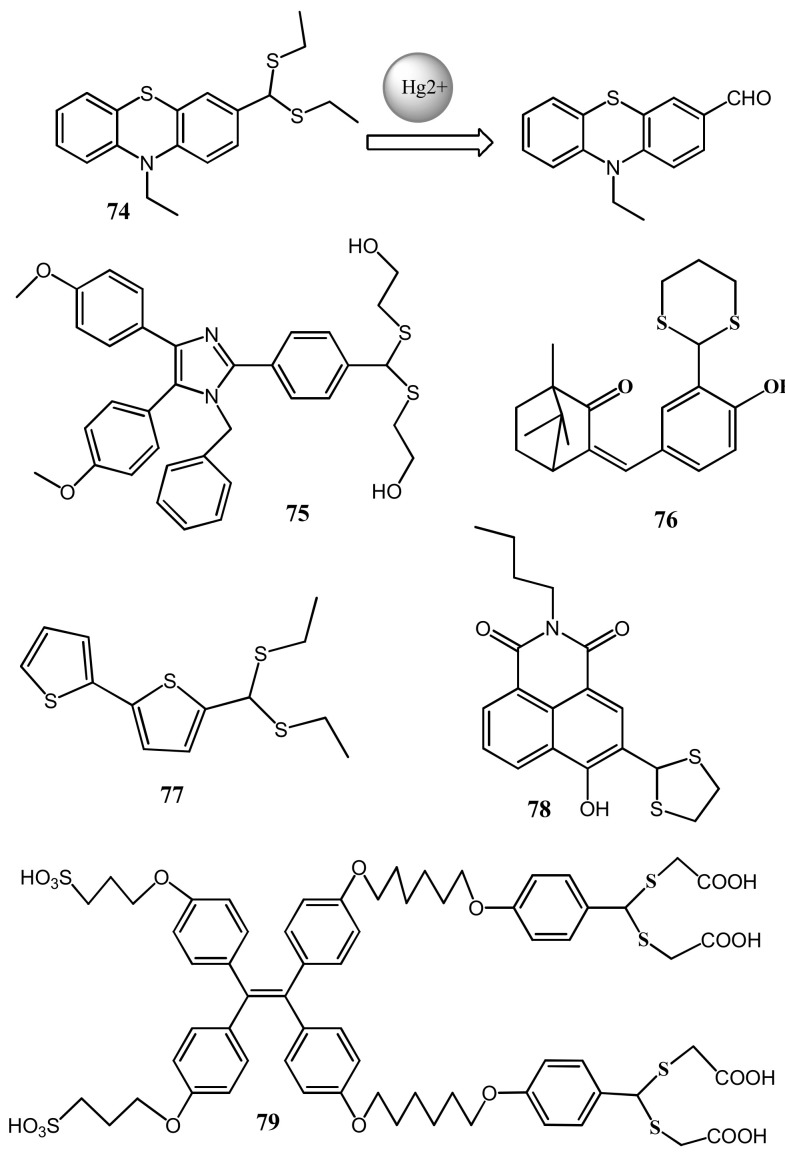
Molecular structures of the sensors **74** to **79**.

**Figure 26 pharmaceuticals-14-00123-f026:**
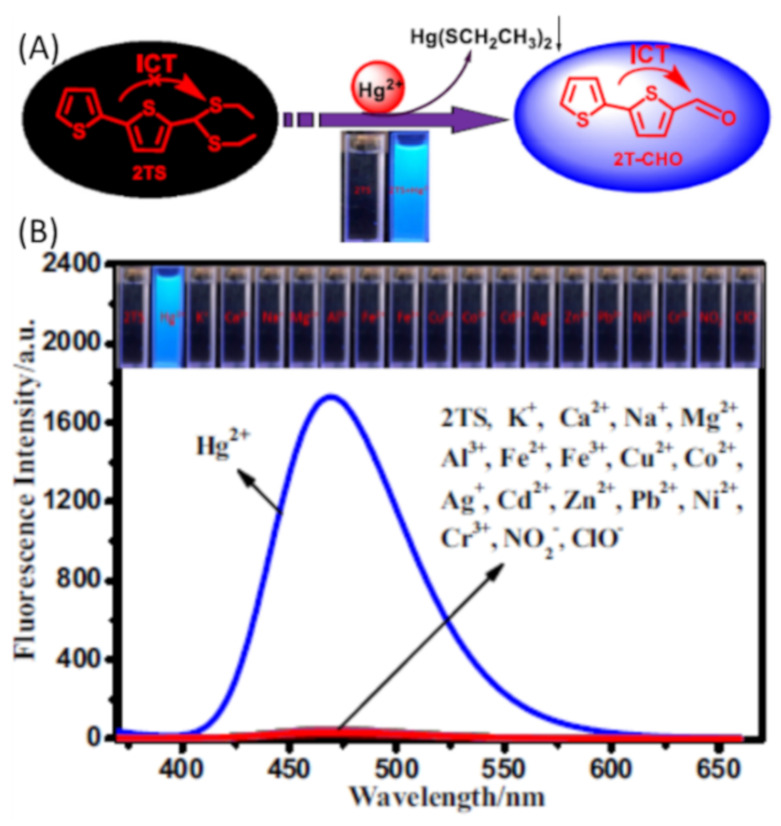
(**A**) Proposed sensing mechanism of **77** (2TS = **77**) for Hg^2+^. (**B**) Fluorescence spectra of **77** (10 μM) after addition of 20 μM of various tested ions in 100% aqueous solution; Inset: fluorimetric responses of **77** (10 μM) in 100% aqueous solution after the addition of various tested ions. Reproduced with permission from [105]. Copyright 2020 Elsevier.

**Figure 27 pharmaceuticals-14-00123-f027:**
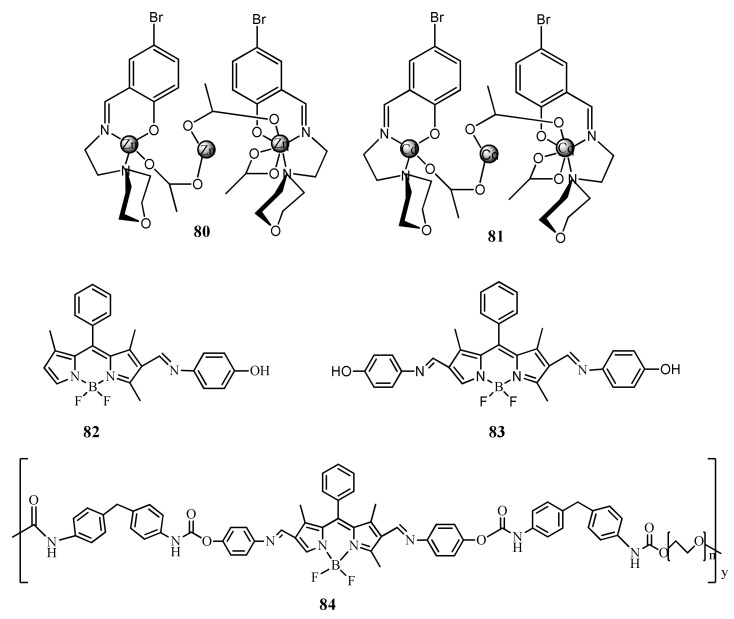
Molecular structures of the sensors **80** to **84**.

**Figure 28 pharmaceuticals-14-00123-f028:**
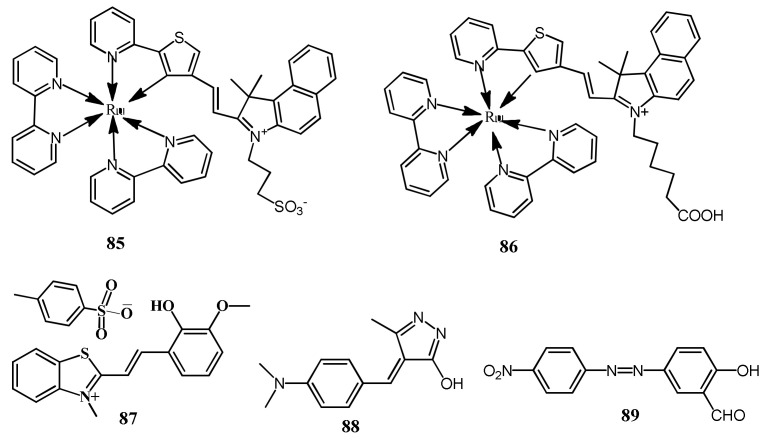
Molecular structures of the sensors **85** to **89**.

**Figure 29 pharmaceuticals-14-00123-f029:**
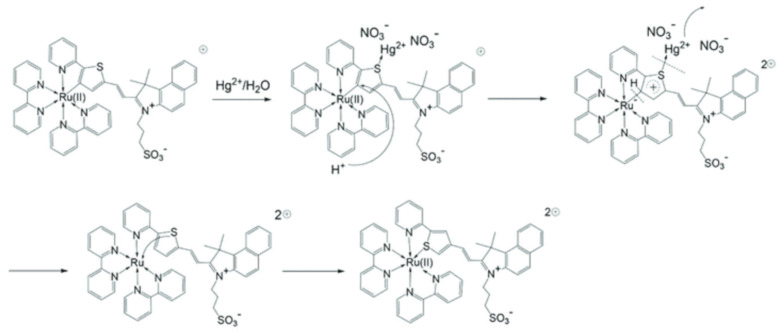
The possible sensing mechanism of 85 with Hg^2+^. Reproduced with permission from [110]. Copyright 2020 The Royal Society of Chemistry.

**Figure 30 pharmaceuticals-14-00123-f030:**
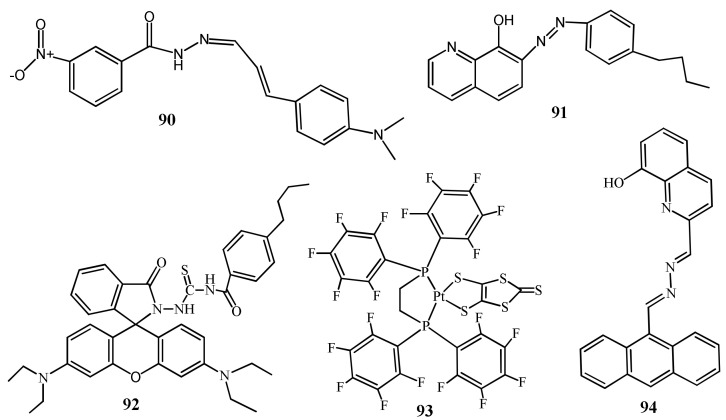
Molecular structures of the sensors **90** to **94**.

**Figure 31 pharmaceuticals-14-00123-f031:**
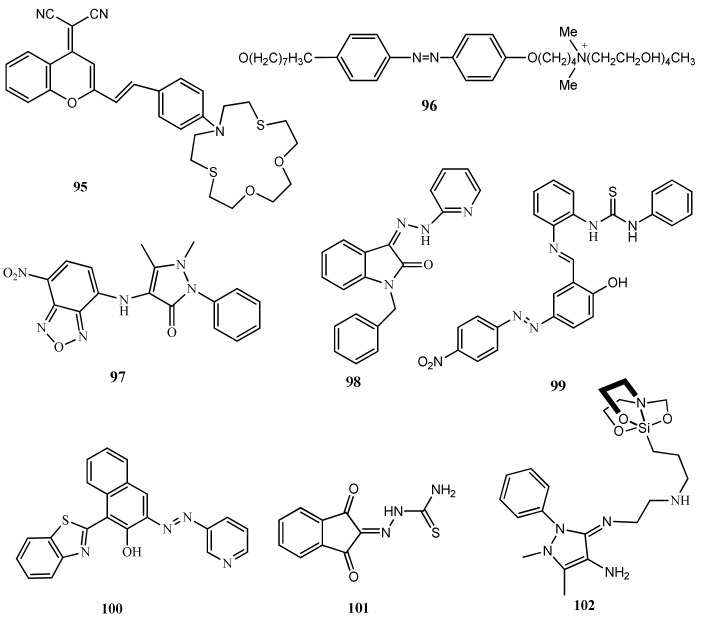
Molecular structures of the sensors **95** to **102**.

**Table 1 pharmaceuticals-14-00123-t001:** Some representative sources of exposure of the different forms of mercury, and the main affected organs [21].

Species	Occupational Exposure	Environmental Exposure	Routes of Exposure	Affected Organs
Elemental mercury, Hg	Chlor-alkali plants, gold extraction, incineration of wastes, coal burning, dental amalgam handling	Dental amalgam	Inhalation	Nervous system
Organic mercury, CH_3_Hg^+^	-	Food (Fish and seafood)	Ingestion	Nervous system
Inorganic mercury, Hg^2+^	-	Medicinal uses, dermatological creams	Ingestion, transdermal	Kidneys

**Table 2 pharmaceuticals-14-00123-t002:** Analytical parameters for the fluorescent sensors **1** to **84**.

Sensors	Medium	λ_exc_/λ_em_ (nm)	LOD	Applications	Ref.
**1**	H_2_O	350/464	0.16 μM	Real water sample analysis	[32]
**2**	H_2_O	500/632	39.2 nM	-	[33]
**3**	H_2_O	316/416	0.1243 µM	Real water sample analysis	[34]
**4**	THF:H_2_O (1:49, *v*/*v*)	380/500	22.8 ppb	-	[35]
**5**	CH_3_CN:H_2_O (1:1, *v*/*v*)	270/380	0.68 μM	Real water analysis and pH sensing	[36]
**6**	EtOH	410/485	172 nM	Living cell imaging	[37]
**7**	CH_3_CN:HEPES buffer (2:8, *v*/*v*)	369/490	1 nM	Live-cell imaging	[38]
**8**	H_2_O	490/574	312 nM	Real water sample analysis	[39]
**9**	Tris-HCL buffer	358/445	4.40 μM	Real water sample analysis and live cell imaging	[40]
**10**	HEPES buffer	330/550	7.59 nM	Real water samples and live cell imaging	[41]
**11**	CH_3_CN:H_2_O (6:4 *v*/*v*)	375/485	1.26 nM	Test paper strip and real water analysis	[42]
**12**	CH_3_CN	420/603	20 ppb	Removal from water	[43]
**13**	DMF	325/463	17 nM	Air and real water samples	[44]
**14**	MeOH	280/436	0.01 μM	-	[45]
**15**	H_2_O	495/521	0.34 μM	Industrial effluents and paper strip	[46]
**16**	H_2_O	430/512	5.02 mM	Real water analysis	[47]
**17**	MeOH:H_2_O (1/4, *v*/*v*)	340/455	3.12 nM	Live cell imaging	[48]
**18**	CH_3_CN	440/492	200 nM	Real water sample analysis	[49]
**19**	DMF:H_2_O (1:1, *v*/*v*)	337/378	16 nM	-	[50]
**20**	H_2_O:DMSO (95:5, *v*/*v*)	360/540	0.51 μM	-	[51]
**21**	H_2_O	285/364,464	7.41 nM	Bioimaging	[52]
**22**	DMSO:HEPES buffer (9:1, *v*/*v*)	355/455	8.12 nM	Live cell imaging	[53]
**23**	H_2_O	359/495	22 nM	Real water analysis and bioimaging	[54]
**24**	MeOH:H_2_O (4:1, *v*/*v*)	305/387	0.28 ppb	Real water analysis	[55]
**25**	DMF:HEPES buffer (1:1, *v*/*v*)	315/495	0.645 µM	Real water analysis and live cell imaging	[56]
**26**	Acetonitrile:HEPES buffer (1:1, *v*/*v*)	401/520	1.01 µM	Real water analysis	[57]
**27**	Acetonitrile:HEPES buffer (1:1, *v*/*v*)	405/525	1.98 µM	-	[57]
**28**	THF:H_2_O (9:1, *v*/*v*)	480/675	13.1 nM	Live cell imaging	[58]
**29**	H_2_O:DMSO (99.6: 0.4, *v*/*v*)	495/543	19.2nM	Test color strips and bio-imaging	[59]
**30**	CH_3_CN/:H_2_O (4:6, *v*/*v*)	520/587	3.9 ppb	Drinking water, live cells and plant tissues	[60]
**31**	CH_3_CN:HEPES (1:9, *v*/*v*)	470/530	27 nM	Live cell imaging	[61]
**32**	MeOH:H_2_O (4:1, *v*/*v*)	360/590	4.8 μM	Test paper strips, bioimaging	[62]
**33**	DMSO:H_2_O mixture	360/453	406 nM	Adsorption of H_2_S	[63]
**34**	DMSO:H_2_O (2:8, *v*/*v*)	-/600	30 nM	Real food samples and live cell imaging	[64]
**35**	HEPES buffer (pH 7.4)	480/518	3 nM	Real water and biological analysis	[65]
**36**	EtOH:H_2_O mixture	366/491	750 nM	Real water sample analysis	[66]
**37**	CH_3_CN/H_2_O (0.2:99.8, *v*/*v*)	366/463	2.1 nM	-	[67]
**38**	DMF:buffer (8:2, *v*/*v*)	315/529	2.40 μM	Live cell imaging	[68]
**39**	DMF:buffer (8:2, *v*/*v*)	470/621	2.86 μM	Live cell imaging	[68]
**40**	DMF:buffer (8:2, *v*/*v*)	470/614	0.22 μM	Live cell imaging	[68]
**41**	H_2_O	480/532	8.619 nM	Real water analysis and live cell imaging	[69]
**42**	PBS buffer	460/515	0.4 nM	Real water analysis and live cell Imaging	[70]
**43**	CH_3_CN:HEPES buffer (1:9, *v*/*v*).	545/580	-	-	[71]
**44**	DMSO	520/606	0.13 µM	INHIBIT logic gate	[72]
**45**	CH_3_CN:HEPES buffer(2:8, *v*/*v*)	580/691	1.5 nM	Living cells imaging	[73]
**45**	EtOH:HEPES buffer (1:1 *v*/*v*)	590/664	1.87 ppb	Real water sample analysis	[74]
**46**	H_2_O	335/574	3.55×10^-13^ mL^−1^	-	[75]
**47**	DMF:H_2_O (7/3, *v*/*v*)	390/504,613	0.2 µM	Living cells	[76]
**48**	CH_3_CN	510/573	28.5 nM	-	[77]
**49**	DMSO:H_2_O (6:4, *v*/*v*)	505/585	16.9 nM	-	[78]
**50**	CH_3_CN	520/585	16 nM	Real water sample analysis	[79]
**51**	DMSO:H_2_O (1:1; *v*/*v*)	500/562	26 nM	Real water sample analysis	[80]
**52**	DMSO:H_2_O (7/3, *v*/*v*)	490/581	14.9 nM	-	[81]
**53**	CH_3_CN:H_2_O (7/3, *v*/*v*)	520/604	0.38 μM	Test color strips and biosensing	[82]
**54**	DMF:Tris-HCl buffer (1:1, *v*/*v*)	562/557	1.61 nM	Bio-sensing and live cell imaging	[83]
**55**	H_2_O:THF (4:1, *v*/*v*)	565/590	16 nM	Live cell imaging	[84]
**56**	DMSO:H_2_O (7:3, *v*/*v*)	480/582	13.4 nM	Test color strips and live cell imaging	[85]
**57**	DMSO:H_2_O (7:3, *v*/*v*)	480/578	15.6 nM	Test color strips and live cell imaging	[85]
**58**	DMSO:H_2_O (7:3, *v*/*v*)	-/560	16.1 nM	Test paper strips and real water analysis	[86]
**59**	H_2_O	-	120 nM	Real water sample analysis	[87]
**60**	HEPES buffer	330/550	23.0 nM	Test color strips and sensing of biothiols	[88]
**61**	HEPES buffer	310/430,548	1.3 nM	Environmental water samples	[89]
**62**	HEPES buffer	330/545	22.65 nM	Biosensing	[90]
**63**	DMSO:HEPES medium (1:99, *v*/*v*)	340/390	14.7 nM	Real water sample analysis	[91]
**64**	EtOH-H_2_O (9:1, *v*/*v*)	500/385,447	0.22 μM	-	[92]
**65**	CH_3_CN:DMSO (99:1, *v*/*v*)	340/395	8.11 nM	IMPLICATION logic gates	[93]
**66**	H_2_O	390/455	7.9	River water and live cell imaging	[94]
**67**	EtOH	300 580	1.10 μM	Test color strips and real water analysis	[95]
**68**	DMSO:H_2_O (1:3, *v*/*v*)	321/444,644	48.79 nM	Real water and beverages samples	[96]
**69**	EtOH:H_2_O (2:8, *v*/*v*)	332/475	146 nM	Real waste water analysis	[97]
**70**	EtOH:HEPES buffer (1:9, *v*/*v*)	450/495,600	1.6 nM	Real water analysis and live cell imaging	[98]
**71**	H_2_O	300/443	39.28 nM	Test color strips and real water analysis	[99]
**72**	THF-H_2_O (1:9, *v*/*v*)	490/667	33 nM	Biological sample and live cell imaging	[100]
**73**	HEPES-DMSO (99:1, *v*/*v*)	325/630	15.1 μM	Bioimaging	[101]
**74**	PBS buffer (pH 7.4)	390/445	21.2 nM	Real water analysis and live cell imaging	[102]
**75**	DMSO:PBS buffer (1:99, *v*/*v*)	380/475	36 nM	Real water analysis and live cell imaging	[103]
**76**	DMSO:PBS buffer (1:99, *v*/*v*)	365/518	19.3 nM	Real sample analysis, test color strips and cell imaging	[104]
**77**	H_2_O	370/470	19 nM	Real water, seafood, human urine samples, test color strips and bio-imaging	[105]
**78**	PBS buffer	414/510	40 nM	Living cell imaging	[106]
**79**	THF:H_2_O (1/99, *v*/*v*)	353/477	20 nM	Test color strips and real water analysis	[107]
**80**	H_2_O	390/461	1.11 μM	-	[108]
**81**	H_2_O	390/464	1.89 μM	-	[108]
**82**	DMF:H_2_O (1:1, *v*/*v*)	330/549	0.21 µM	-	[109]
**83**	DMF:H_2_O (1:1, *v*/*v*)	331/550	0.63 µM	-	[109]
**84**	DMF:H_2_O (1:1, *v*/*v*)	335/559	0.19 µM	-	[109]

**Table 3 pharmaceuticals-14-00123-t003:** Analytical parameters of the colorimetric sensors **85** to **102**.

Sensors	Medium	λ_abs_ (with/without Hg^2+^)	LOD	Applications	Ref.
**85**	Aqueous media	Quenching of bands at 506 and 730 nm	21 nM	-	[110]
**86**	DMSO:HEPES (5:95, *v*/*v*)	Quenching at 503 with the new band formation at 610 nm	53 nM	Polymer coated membrane	[111]
**87**	HEPES buffer	Quenching of bands at 390 and 530 nm	0.27 μM	Test color strips	[112]
**88**	CH_3_CN:H_2_O (7:3, *v*/*v*)	Band at 447 nm shifted to 519 nm	0.473 μM	Test color strips and real water analysis	[113]
**89**	DMSO:H_2_O (4:1 *v*/*v*)	Band at 502 nm shifted to 395 nm	6.1 μM	Cellulose test strips, Logic Gate Operation	[114]
**90**	BufferDMF (98:2, *v*/*v*).	Quenching of bands at 350 and enhancement at 400 nm	0.11 µM	Real water sample analysis	[115]
**91**	Aqueous medium	Enhancement of band at 540 nm	0.100 and 0.180 μg/L	Real water sample analysis	[116]
**92**	Aqueous medium	Enhancement of band at 567 nm	0.22 and 0.61µg/L	Real water sample analysis	[117]
**93**	CH_3_CN:H_2_O (1:1, *v*/*v*)	Band at 448 nm shifted to 523 nm	-	-	[118]
**94**	CH_3_OH:HEPES (7: 3, *v*/*v*)	Band at 414 nm shifted to 498 nm	220 nM	Real water analysis, silica coating and test color strips	[119]
**95**	CH_3_CN:H_2_O (1:1, *v*/*v*).	Band at 517 nm shifted to 415 nm	-	Real water analysis	[120]
**96**	HEPES buffered	Band at 247 nm shifted to 234 nm	40 nM	-	[121]
**97**	CH_3_OH:H_2_O (1: 1, *v*/*v*)	Band at 465 nm shifted to 485 nm	25.7 nM	INHIBIT logic gate	[122]
**98**	MeCN:H_2_O (1:1, *v*/*v*)	Formation of new band at 470 and quenching at 380 nm	0.95 nM	Real water sample analysis	[123]
**99**	DMSO:H_2_O (2:1, *v*/*v*)	Enhancement of band at 280 nm	4.89 μM	Real water sample analysis	[124]
**100**	H_2_O:CH_3_CN (9:1, *v*/*v*)	Quenching of bands at 319 and 380 nm with the formation of new band at 610 nm	8.5 μM	Test color strips and real water sample analysis	[125]
**101**	Aqueous medium	Band at 335 nm shifted to 305 nm	1 μM	Real time application	[126]
**102**	DMSO:H_2_O (8:2, *v*/*v*)	Band at 290 nm shifted and two new bands formed at 255 and 292 nm	0.10 mM	-	[127]

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
