# Peer review of "Mercury Toxicity and Detection Using Chromo-Fluorogenic Chemosensors"

_pharmaceuticals, 2021, doi:10.3390/ph14020123_

Round 1

Reviewer 1 Report

The review manuscript by Bhardwaj et al. mainly summarized mercury sensors published in 2020. The review is very inclusive, but the overall manuscript logic and organization makes it quite hard to follow. Here are some detailed suggestions:

  1. In order to clearly state the significance and necessity to develop mercury sensors, the authors must do a better job explaining the background. There is some important information is missing, such as what is the typical concentration range of mercury for each type of exposure? What is the typical concentration range of mercury polluted water, food, or air? Such information is very limited in the background part, a short description of occupation exposure concentration only. This part should be expanded and better stated.
  2. Because the authors focused more on the toxicity of mercury to humans, it would be better to include some biological background regarding mercury toxicity. For example, what happens when Hg (different forms) enters human cells? What is the toxicology molecular mechanism? How does cellular detoxification work? When will the innate detoxification mechanism such as glutathione and cyp enzymes be overwhelmed by mercury and how? What is the typical toxicity concentration that leads to acute, long-term, or severe damage?
  3. Table 1 looks incomplete and disorganized. The authors stated that organic mercury mostly comes from fish and food and damage nerve systems, but the table says different. Also, the table and related information lacks literature citations and is overall not clear. There have been reports that organic mercury is the most common metabolized end product that actually causes severe irreversible nerve injury. The authors should expand more on organic mercury and how the metabolism of human cells could contribute to the overall toxic effect.
  4. There should be more subtitles for each section, such as ‘turn-off’ fluorescent sensors, ‘turn-on’ fluorescent sensors. This will greatly help readers to navigate through the review. I also suggest adding some short introductory paragraphs under each sub-title and state the authors' opinion on the advantages and disadvantages of each category.
  5. I suggest reorganizing within each sub-section based on sensing mechanism, as it is the most important aspect of any sensor. The fluorophore can be easily switched if the core sensing mechanism is good. The mechanism also guides future sensor development and should be discussed further and compare with existing widely used mechanisms such as general coordination with thiol groups.
  6. I suggest summarizing all sensors into tables for better reading and comparisons. The authors can list the key elements of each sensor including LOD, selectivity (tested interfering compounds), kinetics, solvent composition, optical properties such as wavelength, quantum yields, pH ranges, application scenarios, etc. Tables for each subsection will greatly facilitate comparisons between methods. It is also advised to add a few classic sensors for comparison to show how the field improved over years.
  7. There should be highlighted sensors and future advice based on the highlights. Merely listing all data does not summarize the field and help improve future development. The authors must add the opinion and thoughts in the comments sections either for each subsection or overall to highlight how existing sensors help future development and where are the most urgent needs.
  8. I suggest removing all COVID-19 related statements, especially in the abstract part. This is not relevant at all and may cause confusion. The review only talks about sensors in terms of their chemistry, and none of the reported sensors have commented on COVID-19 related applications.
  9. The writing style may need some further professional editing.
  10. Page 2, line 68 there is a typo of 'xx century'.

Author Response

The review manuscript by Bhardwaj et al. mainly summarized mercury sensors published in 2020. The review is very inclusive, but the overall manuscript logic and organization makes it quite hard to follow. Here are some detailed suggestions:

Reply: thank you for the valuable suggestions to improve our review. We have acted according to your suggestions. The changes made in the revised manuscript are mentioned below point by point:

  1. In order to clearly state the significance and necessity to develop mercury sensors, the authors must do a better job explaining the background. There is some important information is missing, such as what is the typical concentration range of mercury for each type of exposure? What is the typical concentration range of mercury polluted water, food, or air? Such information is very limited in the background part, a short description of occupation exposure concentration only. This part should be expanded and better stated.

Reply: The concentration range of mercury required for the design and development of chemosensors was discussed briefly in the introduction section.

  1. Because the authors focused more on the toxicity of mercury to humans, it would be better to include some biological background regarding mercury toxicity. For example, what happens when Hg (different forms) enters human cells? What is the toxicology molecular mechanism? How does cellular detoxification work? When will the innate detoxification mechanism such as glutathione and cyp enzymes be overwhelmed by mercury and how? What is the typical toxicity concentration that leads to acute, long-term, or severe damage?

Reply: Thank you for the suggestion. We have added some text in introduction sections along with a reference (Ref. 7) on ‘mercury toxicity model’ for the step wise effects of mercury on human health.

  1. Table 1 looks incomplete and disorganized. The authors stated that organic mercury mostly comes from fish and food and damage nerve systems, but the table says different. Also, the table and related information lacks literature citations and is overall not clear. There have been reports that organic mercury is the most common metabolized end product that actually causes severe irreversible nerve injury. The authors should expand more on organic mercury and how the metabolism of human cells could contribute to the overall toxic effect.

Reply: Table 2 was modified and supported by citing a reference (Ref. 21).

  1. There should be more subtitles for each section, such as ‘turn-off’ fluorescent sensors, ‘turn-on’ fluorescent sensors. This will greatly help readers to navigate through the review. I also suggest adding some short introductory paragraphs under each sub-title and state the authors' opinion on the advantages and disadvantages of each category.

Reply: The advantages and disadvantages of various fluorescence sensing approached were briefly mentioned.

  1. I suggest reorganizing within each sub-section based on sensing mechanism, as it is the most important aspect of any sensor. The fluorophore can be easily switched if the core sensing mechanism is good. The mechanism also guides future sensor development and should be discussed further and compare with existing widely used mechanisms such as general coordination with thiol groups.

Reply: The sensors working with similar sensing mechanism are explained by adding additional information.

  1. I suggest summarizing all sensors into tables for better reading and comparisons. The authors can list the key elements of each sensor including LOD, selectivity (tested interfering compounds), kinetics, solvent composition, optical properties such as wavelength, quantum yields, pH ranges, application scenarios, etc. Tables for each subsection will greatly facilitate comparisons between methods. It is also advised to add a few classic sensors for comparison to show how the field improved over years.

Reply: Two new tables were added in the revised table to add the important analytical parameters of the fluorescent and colorimetric sensors.

  1. There should be highlighted sensors and future advice based on the highlights. Merely listing all data does not summarize the field and help improve future development. The authors must add the opinion and thoughts in the comments sections either for each subsection or overall to highlight how existing sensors help future development and where are the most urgent needs.

Reply: The conclusion section was revised and discussion was made on future scope and research.

  1. I suggest removing all COVID-19 related statements, especially in the abstract part. This is not relevant at all and may cause confusion. The review only talks about sensors in terms of their chemistry, and none of the reported sensors have commented on COVID-19 related applications.

Reply: The discussion on COVID-19 was removed from abstract and introduction section.

  1. The writing style may need some further professional editing.

Reply: The writing style was improved.

  1. Page 2, line 68 there is a typo of 'xx century'.

Reply: The ‘XX century’ was corrected as ‘20th century’.

Reviewer 2 Report

This paper by V. Bhardwaj et al. constitutes a quite interesting review article about chromo-fluorogenic sensors for Hg2+ ions reported in the previous year. The review contains a lot of sufficiently detailed information about various sensors what is demonstrated by a considerable list of references. The quality of the text is rather good. This is important, especially for review articles, to make sure the manuscript is more easily readable by target audience. In my opinion the manuscript with no doubt merits to be published in Pharmaceuticals after revision.

  1. In my opinion, it would be a good idea to make “a list” of the most important things (for example advantages and disadvantages offered by fluorogenic systems) about the reported sensors as a table at the end of the manuscript. Such a tabular presentation is often very useful and allows for an easy comparison.
  2. Line 208: the term “Table 5 showed…” makes a fuss. It is not clear if table 5 is in the manuscript or in appropriate reference. It corresponds to the whole manuscript and should be changed or at least explained.

Author Response

This paper by V. Bhardwaj et al. constitutes a quite interesting review article about chromo-fluorogenic sensors for Hg2+ ions reported in the previous year. The review contains a lot of sufficiently detailed information about various sensors what is demonstrated by a considerable list of references. The quality of the text is rather good. This is important, especially for review articles, to make sure the manuscript is more easily readable by target audience. In my opinion the manuscript with no doubt merits to be published in Pharmaceuticals after revision.

Reply: Thank you for the valuable suggestion. We have acted and revised our manuscript according to your suggestions. Please see below the point by point explanation of the changes made in the revised manuscript.

  1. In my opinion, it would be a good idea to make “a list” of the most important things (for example advantages and disadvantages offered by fluorogenic systems) about the reported sensors as a table at the end of the manuscript. Such a tabular presentation is often very useful and allows for an easy comparison.

Reply: Two new tables were added in the revised table to add the important analytical parameters of the fluorescent and colorimetric sensors.

  1. Line 208: the term “Table 5 showed…” makes a fuss. It is not clear if table 5 is in the manuscript or in appropriate reference. It corresponds to the whole manuscript and should be changed or at least explained.

Reply: The revised manuscript contains only three tables. We have edited the manuscript as per reviewer suggestion.

Reviewer 3 Report

Manuscript title: Mercury toxicity and detection using chromo-fluorogenic Chemosensors

In this work, Sahoo and colleagues provide a very good review on the Hg toxicity and the methods for the detection. The Hg toxicity was first discussed, followed by the comprehensive review on the small molecular probes for Hg detection by colorimetric and fluorescent sensors. This is an excellent review article. The structure of this work is very clear, will definitely attract broad interests to readers of pharmaceuticals. The manuscript is recommended for publishing after addressing following minor questions:

  1. Cause of COVID-19 was highlighted in the abstract section, while COVID-19 was not discussed in the main article. The review suggests removing this description from the abstract to avoid potential confusion to readers.
  2. Chemical structures of colorimetric and fluorescent probes were well presented in the manuscript. Response mechanisms of the probes for Hg detection is suggested to be highlighted (maybe a scheme to highlight the response mechanism will be helpful).
  3. Metal complexes-based probes for Hg detection, for example, Tb-complex for time-gated luminescence detection of Hg (Journal of fluorescence 2012, 22 (1), 261-267) is suggested to be discussed.
  4. In the conclusion section, the application of Hg probes in real-world samples detection is suggested to be discussed in a few sentences.

Author Response

In this work, Sahoo and colleagues provide a very good review on the Hg toxicity and the methods for the detection. The Hg toxicity was first discussed, followed by the comprehensive review on the small molecular probes for Hg detection by colorimetric and fluorescent sensors. This is an excellent review article. The structure of this work is very clear, will definitely attract broad interests to readers of pharmaceuticals. The manuscript is recommended for publishing after addressing following minor questions:

Reply: Thank you for the valuable suggestion. We have acted and revised our manuscript according to your suggestions. Please see below the point by point explanation of the changes made in the revised manuscript.

  1. Cause of COVID-19 was highlighted in the abstract section, while COVID-19 was not discussed in the main article. The review suggests removing this description from the abstract to avoid potential confusion to readers.

Reply: The discussion on COVID-19 was removed from abstract and introduction section.

  1. Chemical structures of colorimetric and fluorescent probes were well presented in the manuscript. Response mechanisms of the probes for Hg detection is suggested to be highlighted (maybe a scheme to highlight the response mechanism will be helpful).

Reply: Some scheme and figures were added in the revised manuscript to visualize the sensing mechanism.

  1. Metal complexes-based probes for Hg detection, for example, Tb-complex for time-gated luminescence detection of Hg (Journal of fluorescence 2012, 22 (1), 261-267) is suggested to be discussed.

Reply: Metal-complex based sensing approach was discussed.

  1. In the conclusion section, the application of Hg probes in real-world samples detection is suggested to be discussed in a few sentences.

Reply: Real-world applications of the sensors were discussed in the conclusion section.

Round 2

Reviewer 2 Report

In my opinion the revised version of the manuscript merits to be published in Pharmaceuticals journal.

Author Response

Thank you for accepting our manuscript.